# How Temporal Unrolling Supports Neural Physics Simulators

## Abstract

Unrolling training trajectories over time strongly influences the inference accuracy of neural network-augmented physics simulators. We analyze these effects by studying three variants of training neural networks on discrete ground truth trajectories. In addition to commonly used one-step setups and fully differentiable unrolling, we include a third, less widely used variant: unrolling without temporal gradients. Comparing networks trained with these three modalities makes it possible to disentangle the two dominant effects of unrolling, *training distribution shift* and *long-term gradients*. We present a detailed benchmark across physical systems, network sizes, network architectures, training setups, and test scenarios. It provides an empirical basis for our main findings: Fully differentiable setups perform best across most tests, yielding an improvement of 38% on average. Nevertheless, the accuracy of unrolling without temporal gradients comes comparatively close with 23%. These results motivate integrating non-differentiable numerical simulators into training setups even if full differentiability is unavailable. Furthermore, we empirically show that these behaviors are invariant to changes in the underlying physical system, the network architecture and size, and the numerical scheme.

## 1 Introduction

Our understanding of physical systems relies on capturing their dynamics in mathematical models, often representing them with a partial differential equation (PDE). Forecasting the behavior with these models thus involves the notoriously costly and difficult task of solving the PDE. By aiming to increase simulator efficiencies, machine learning was successfully deployed to augment traditional numerical methods for solving these equations. Common areas of research are network architectures (Sanchez-Gonzalez et al., 2020; Li et al., 2020; Geneva & Zabaras, 2020; Ummenhofer et al., 2019), reduced order representations (Lusch et al., 2018; Wiewel et al., 2019; Eivazi et al., 2021; Brunton et al., 2021; Wu et al., 2022), and training methods (Um et al., 2020; Sirignano et al., 2020; MacArt et al., 2021; Brandstetter et al., 2022). Previous studies were further motivated by performance and accuracy boosts offered by these methods, especially on GPU architectures (Beck & Kurz, 2021).

For hybrid simulators, neural network components are intended to run along the numerics at simulation time. Thus, long-term autoregressive stability and accuracy are required. In this setting, the most straightforward setups, e.g. learning single update steps in a fully supervised manner, are not ideal. Instead, integrating the numerical solver into the gradient backpropagation for machine learning is a capable alternative (Hu et al., 2019; Holl et al., 2020b). Unrolling multiple simulator steps during training drastically improved results compared to previously studied one-step training based on non-differentiable simulators (Kochkov et al., 2021). This success is usually attributed to two fundamental benefits of unrolling: (1) being aware of *data shift*, and (2) using information about the *long-term interactions* of the underlying physical system (Um et al., 2020; Brandstetter et al., 2022). Data shift during inference is a natural consequence of inferring long-term trajectories (Wiles et al., 2021). For chaotic systems, we can go one step further: When the learned system does not yield dynamics identical to the ground truth, data sampled from this ground truth distribution will not fully explore the learned dynamics. This data shift can be seen as the difference between the learned dynamics attractor and the ground truth attractor. In contrast, unrolling fully exposes the learned attractor as the unrolled horizon grows. See Appendix B for theoretical details.

In principle, this would motivate ever longer horizons, as illustrated in Figure 1. However, obtaining useful gradients through unrolled chaotic systems limits this horizon in practice. The differentiability of numerical solvers is crucial to backpropagate through solver-network chains. Chaotic effects (Mikhaeil et al., 2022) or long recurrent network chains (Pascanu et al., 2013) eventually lead to gradient divergence in these backpropagation chains. As an alternative, we study non-differentiable unrolling as a new variant. This training modality has a high practical relevance, as it can be realized by integrating existing, non-differentiable solvers into a deep learning framework. As a natural consequence, gradients diverge from the loss landscape for long unrollings when using this variant. Figure 1 illustrates these gradient inaccuracies for both training modalities, while a theoretical analysis is given in Appendix B. Our work provides new insights about advantages and disadvantages of this previously unused, but practically important variant. We rigorously analyze these setups using standardized solver and network

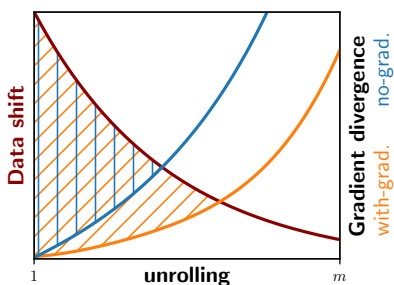

Figure 1: Illustration of data shift and gradient divergence over unrolling; gradients of non-differentiable unrolling diverge from the true gradient landscape (blue), gradients of differentiable simulators are prone to explosions over long horizons (orange); shaded areas mark potential benefits over one-step learning, with-gradient is best.

architectures. This allows disentangling the effects of training data shift introduced by forward unrolling and accurate long-term gradients calculated by backpropagation. We test our findings on various physical systems with multiple network architectures, including convolutional and graph networks. Five recommendations for unrolled neural simulators training are derived from our results:

**(I) Low-dimensional problems match:** Our results translate between physical systems. Provided the systems are from a similar domain, i.e. we primarily consider turbulent chaotic systems, high-level training behaviors translate between physical systems. This validates the approach common in other papers, where broad studies were conducted on low-dimensional systems and only tested on the high-dimensional target.

**(II) Non-differentiable unrolling offers great value:** The mitigating effects on data shift introduced by unrolling can boost model performance. An unrolled but non-differentiable chain can increase performance by 23% on average. While the computation costs per optimization step increase with unrolling, the costs are still lower than full differentiable setups. In addition, non-differentiable neural hybrid simulators do not require a new implementation of existing numerical solvers and thus represent an attractive alternative for fields with large traditional code bases.

**(III) Differentiable unrolling delivers the best accuracy:** When other hyperparameters are fixed, a differentiable unrolling strategy consistently outperforms other setups. At the cost of increased software engineering efforts, differentiable unrolling outperforms its one-step counterpart by 38 percent on average. Long-term gradients prove to be especially important for hybrid setups where neural networks correct numerical solvers.

**(IV) Curriculums are necessary:** The training of unrolled setups is non-trivial and sensitive to hyperparameters. Reliable training setups utilize a curriculum where the unrolled number of steps slowly increases. The learning rate needs to be adjusted to keep the amplitude of the gradient feedback at a stable level.

**(V) Parameter count matters:** Network size remains the predominant factor regarding accuracy on inter- and extrapolative tests. While large networks generally give best results, this somewhat unsurprising finding comes with a caveat: Most neural PDE simulators directly compete with numerical solvers. Thus, inference performance becomes critical. Our broad evaluation provides convergence rates that show suboptimal scaling compared to numerical solvers, highlighting the fundamental importance of resource-efficient architectures. Consequently, medium-sized networks are preferable, and successful unrolling is crucial to obtain the best performance for a chosen network size.

Our benchmark differentiates itself from previous work with its task and architecture-agnostic stance. Combined with the large scale of evaluations across more than three thousand models, this allows for extracting general trends and the aforementioned set of recommendations. Source code and training data will be made available upon acceptance.

## 2 RELATED WORK

**Data-driven PDE solvers:** Machine learning based simulators aim to address the limitations of classical PDE solvers in challenging scenarios with complex dynamics (Frank et al., 2020; von Rueden et al., 2020; Brunton et al., 2020). While physics-informed networks have gained popularity in continous PDE modeling (Raissi et al., 2017; Duraisamy et al., 2019), many learned approaches work in a purely data-driven manner on discrete trajectories. Several of these use advanced network architectures to model the time-evolution of the PDE, such as graph networks (Pfaff et al., 2020; Sanchez-Gonzalez et al., 2020), problem tailored architectures (Wang et al., 2020; Stachenfeld et al., 2021), bayesian networks (Yang et al., 2021), transformer models (Han et al., 2021; Geneva & Zabaras, 2022a; Li et al., 2022), or lately diffusion models (Lienen et al., 2023; Lippe et al., 2023; Kohl et al., 2023). Variations of these approaches compute the evolution in an encoded latent space (Wiewel et al., 2019; Geneva & Zabaras, 2020; 2022b; Brunton et al., 2021; Wu et al., 2022).

**Unrolled training:** Fitting an unrolled ground truth trajectory is frequently used for training autoregressive methods. This can be done with unrolled architectures that solely rely on networks (Geneva & Zabaras, 2020), or for networks that correct solvers (Um et al., 2020; Kochkov et al., 2021; MacArt et al., 2021; Melchers et al., 2023). For the latter, differentiable or adjoint solvers allow backpropagation of gradients through the entire chain (Hu et al., 2019; Sirignano et al., 2020; Holl et al., 2020a). The studies building on these solvers report improved network performance for longer optimization horizons. However, differentiability is rarely satisfied in existing code bases. A resulting open question is how much the numerical solver's differentiability assists in training accurate networks. As the introduction mentions, two properties of differentiable unrolling affect the training procedure. The *data shift* moves the observed training data closer toward realistic inference scenarios (Wiles et al., 2021; Wang et al., 2022). For instance, Brandstetter et al. (2022); Prantl et al. (2022) proposed variations of classic truncation (Sutskever, 2013), and reported a positive effect on the network performance, where "warm-up" steps without contribution to the learning signal stabilize the trained networks. The differentiability of the numerical solver introduces the second property, which allows the propagation of gradients through time evolutions. The resulting loss landscape better approximates temporal extrapolation, which benefits model performance (Um et al., 2020; Sirignano et al., 2020; Kochkov et al., 2021). Recently, possible downsides of differentiable unrolling concerning gradient stability were investigated. While a vanishing/exploding effect is well known for recurrent networks, it is only sparsely studied for hybrid approaches. Mikhaeil et al. (2022) found that the stability of the backpropagation gradients aligns with the Lyapunov time of the physical system for recurrent prediction networks. List et al. (2022) have proposed to cut the backpropagation chain into individual sequences.

**Benchmarks and datasets:** Several benchmarks and datasets have been published to increase comparability and promote standardization of machine-learned PDE simulators, especially in fluid mechanics. A dataset of measured real-world smoke clouds can be found in (Eckert et al., 2019). Bonnet et al. (2022) provide a high-fidelty Reynolds-averaged Navier-Stokes dataset. Furthermore, more specialized datasets like (Xian et al., 2022) focus on fluid manipulation and robotics. Janny et al. (2023) have generated a dataset of large-scale 2D fluid flows on non-uniform meshes and established a benchmark for transformer models. As we focus on tasks that correct solvers, we evaluate our models on frameworks with differentiable solvers (Holl et al., 2020a; Um et al., 2020).

## 3 UNROLLING PDE EVOLUTIONS

Unrolling is a common strategy for learning time sequences. A first important distinction is whether the task at hand is a *prediction* or a *correction* task. Additionally, we formally introduce the differences between *non-differentiable* and *differentiable* unrolling in gradient calculation.

**Evolving partial differential equations with neural networks:** Let us consider the general formulation of a PDE in the form of $\partial \mathbf{u}/\partial t = \mathcal{F}(\mathbf{u}, \nabla \mathbf{u}, \nabla^2 \mathbf{u}, \dots)$, with $\mathbf{u}$ representing the field variables of the physical system. We study *Neural Simulator* architectures that use neural networks for evolving discretized forms of PDEs. They are autoregressively applied to generate trajectories of discrete $\mathbf{u}^t$. *Prediction* setups fully rely on a neural network to calculate the next timestep as $\mathbf{u}^{t+1} = f_\theta(\mathbf{u}^t)$, where $\theta$ represents the network parameters. In a prediction configuration, the network fully replaces a numerical solver with the goal to improve its accuracy and performance. Similarly, *correction* setups are also concerned with time-evolving a discretized PDE but additionally

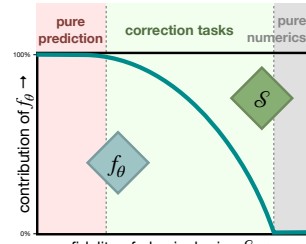

Figure 2: Left: Illustration of unrolling for a horizon $m = 2$ for correction and prediction chains, including gradient flow in the backward pass for non-differentiable (blue, NOG) and differentiable (orange, WIG) setups, note how the gradients do not flow through a time step for NOG; Right: Contributions of networks and numerical solvers in prediction and correction paradigms

include a physical prior $\mathcal{S}$ that approximates the solution. The network then *corrects* this approximation such that $\mathbf{u}^{t+1} = f_\theta(\mathcal{S}(\mathbf{u}^t))$. These architectures represent a *hybrid* between numerical and neural architectures, potentially combining advantages of both domains. Crucially, inputs and outputs of the networks live in the space of the solution vector field, both for predictive and corrective setups. Thus, we use different instances of the same architecture in both setups.

**Training neural networks to solve PDEs:** The network models are trained to represent the underlying PDE by fitting a dataset of ground truth trajectories $\tilde{U} = [\tilde{\mathbf{u}}^0 \dots \tilde{\mathbf{u}}^N]$. The general training procedure is given by as

$$\theta^* = \arg\min_\theta \left[ \sum_{i=1}^{N-m} \sum_{s=1}^{m} \mathcal{L}_2\big(\tilde{\mathbf{u}}^{i+s\tau}, g^s(\mathbf{u}^i)\big) \right], \tag{1}$$

where $g^s$ represents the recurrent application of $s$ simulator steps. Furthermore, $g^{s+1} = f_\theta(\mathcal{S}(g^s))$ for correction and $g^{s+1} = f_\theta(g^s)$ for prediction with $g^0 = \mathbf{u}^i$. $\tau$ accounts for the relative timestep between ground truth and predicted trajectories.

We use three different methods of generating and propagating optimization gradients for network training. The simplest approach utilizes a one-step (ONE) evolution in the forward process with $m = 1$. An extension of the ONE setup uses unrolled trajectories during training. The with-gradient (WIG) setup differentiates this unrolled trajectory in the backward pass. The neural network updates consist of accumulated backpropagation paths from each unrolled state to all previous applications of $f_\theta$. In contrast, the no-gradient (NOG) setup assumes that no differentiable solver is available. This is the case in most scientific computing codebases. The NOG setup is a learning approach that only requires interfacing these codebases with the network at training

Table 1: Calculated gradients with unrolled step $s$; for correction: $\frac{\partial g^{s+1}}{\partial g^s} = \frac{\partial f_\theta^s}{\partial \mathcal{S}^s} \frac{\partial \mathcal{S}^s}{\partial g^s}$; for prediction: $\frac{\partial g^{s+1}}{\partial g^s} = \frac{\partial f_\theta^s}{\partial g^s}$

|  | $\mathcal{L}$ | $\partial\mathcal{L}/\partial\theta$ |
|---|---|---|
| ONE | $\mathcal{L}_2^1$ | $\dfrac{\partial\mathcal{L}_2^1}{\partial f_\theta^1}\dfrac{\partial f_\theta^1}{\partial\theta}$ |
| NOG | $\sum_s \mathcal{L}_2^s$ | $\sum_s \dfrac{\partial\mathcal{L}_2^s}{\partial f_\theta^s}\dfrac{\partial f_\theta^s}{\partial\theta}$ |
| WIG | $\sum_s \mathcal{L}_2^s$ | $\sum_s \sum_{B=1}^{s} \dfrac{\partial\mathcal{L}_2^s}{\partial g^s}\dfrac{\partial g^s}{\partial g^B}\dfrac{\partial g^B}{\partial\theta}$ |

time. Herein, no gradients flow from one recurrent application back to the previous one, and hence the loss is individually computed for each unrolled step. Figure 2 contains a visualization including the backpropagation flow through an unrolled chain. The gradient calculations for our three setups are further denoted in table 1. Further detail of the gradient calculations is found in Appendix A.

## 4 PHYSICAL SYSTEMS AND ARCHITECTURES

Four physical systems were used for our learning tests. All systems are parameterized to exhibit varying behavior, and each test set contains unseen values inside and outside the range of the training data set. Further details of the systems and architectures are provided in Appendix C.

**Kuramoto-Sivashinsky (KS):** This equation is a fourth-order chaotic PDE. The domain size, which leads to more chaotic behavior and shorter Lyapunov times for larger values (Edson et al.,

Figure 3: Visualizations of our physical systems, from left to right: KS equation state, KOLM vorticity field, WAKE vorticity field, AERO Mach numbers

2019), was varied across training and test data sets. The ground truth solver uses an exponential RK2 integrator (Cox & Matthews, 2002), the base solver for correction resorts to a first order version that diverges within 14 steps on average. We base most of our empirical tests on this KS case since it combines challenging learning tasks with a small computational footprint.

**Wake Flow (WAKE):** Our second system is a two-dimensional flow around a cylinder, governed by the incompressible Navier-Stokes equations. The dataset consists of Karman-Vortex streets with varying Reynolds numbers, and is simulated using a Chorin projection with operator splitting (Ferziger et al., 2019). The training data is generated with second order advection, while learning tasks use a truncated spatial resolution ($4\times$), and first order advection as correction base solver.

**Kolmogorov Flow (KOLM):** Thirdly, we study a periodic two-dimensional Kolmogorov flow (Givental et al., 2009) also following the incompressible Navier-Stokes equations. We use a more involved semi-implicit numerical scheme (PISO by (Issa, 1986)) for this setup. Similarly to the WAKE system, the KOLM learning cases are based on a spatiotemporal resolution truncation with a ratio of $4\times$ in both space and time, and the Reynolds number is varied across training and test cases.

**Compressible Aerofoil flow (AERO):** In the final physical system, we conduct an investigation into compressible flow around an aerofoil, a task centered on pure prediction. We utilize the open-source structured-grid code CFL3D (Rumsey et al., 1997; Rumsey, 2010) to solve the compressible Navier-Stokes equations, thus generating our ground truth dataset. Here, we vary the Mach number while keeping the Reynolds number constant.

**Neural Network Architectures:** The following benchmarking results primarily employ two popular architectures: a message-passing graph network (Scarselli et al., 2008; Sanchez-Gonzalez et al., 2020) and a convolutional ResNet (He et al., 2016). Both use residual blocks with skip-connections. The number of layers varies in line with classical ResNet architectures to obtain network sizes spanning several orders of magnitude. Additional U-net results are reported in Appendix D.

## 5 RESULTS

Our results compare the three training methods ONE, NOG, and WIG in various scenarios. The figure titles mark the physical system, whilst subscripts represent the network architecture (*graph*: graph network, *conv*: convolutional ResNet), and superscripts differentiate between correction (*corr*) and prediction (*pred*) tasks. For each test, we train multiple models (typically 8 to 20) for each setup that differ only in terms of initialization (i.e. random seed). The evaluation metrics were applied independently for outputs generated by each of the models, which are displayed in terms of mean and standard deviations below. This indicates the expected performance of a training setup and the reliability of obtaining this performance. A Welch's test for statistical significance was performed for the resulting test distributions, and p-values can be found alongside all tabulated data in Appendix F. We focus on the $\mathcal{L}_2$ loss as a evaluation metric for our tests. Other test metrics lead to similar conclusions and are shown in Appendix D.

### 5.1 DISTRIBUTION SHIFT AND LONG-TERM GRADIENTS

**Agnosticisms:** We first focus on establishing a common ground between the different variants by focusing on correction tasks for the physical systems. Training over 400 models with different architectures, initialization, and parameter counts, as shown in figure 4, reveals a first set of fundamental

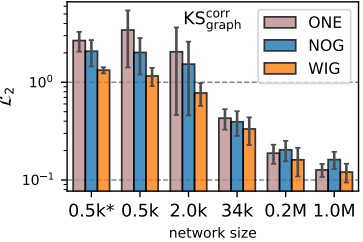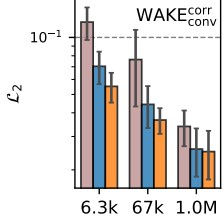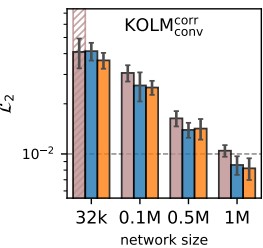

Figure 4: Inference accuracy measured in $\mathcal{L}_2$ for correction setups on KS, KOLM, and WAKE systems; displayed models were trained with ONE(brown), NOG(blue), WIG(orange); across network architectures (graph networks for KS, conv-nets otherwise), and network sizes WIG has lowest errors; one 32k ONE model diverged in the KOLM case that NOG and WIG kept stable

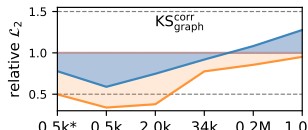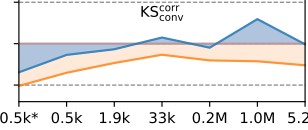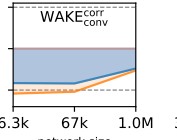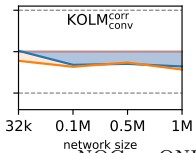

Figure 5: $\mathcal{L}_2$ error of NOG and WIG setups relative to the error achieved by ONE (i.e. $\mathcal{L}_2^{\text{NOG}}/\mathcal{L}_2^{\text{ONE}}$ and $\mathcal{L}_2^{\text{WIG}}/\mathcal{L}_2^{\text{ONE}}$); WIG training reliably produces more accurate results than NOG and ONE

properties of training via unrolling: it reduces inference errors for graph- and convolutional networks, despite changes in dimensionality, the order of the PDE, and the baseline solver architecture. For the vast majority of tests, models trained with unrolling outperform the corresponding one-step baselines. Unrolling is also versatile concerning the type of the modeled correction, as networks were tasked to either learn convergence order truncations with graph networks (KS), or spatial grid coarsening errors via CNNs (KOLM, WAKE). These findings are not new (Bar-Sinai et al., 2019; Um et al., 2020), but confirm that our setup matches previous work. They already motivate recommendation (I), i.e. that testing unrolled architectures can be performed efficiently on cheaper low-dimensional problems, such as the KS system.

**Disentangling Contributions:** Next, we investigate the effect of backpropagating gradients in the unrolled chain of NN and simulator operations. The non-differentiable NOG setup already addresses the data shift problem, as it exposes the learned dynamics' attractor at training time. The inference errors are visualized in figure 4. On average, training with NOG over ONE training yields an error reduction of 23%, in line with recommendation II. For large architectures, inference accuracy increases and learned and ground truth systems become more alike. In these cases, their attractors are similar, and unrolling is less crucial to expose the learned attractor. However, NOG training still uses crude gradient approximations, which makes this variant fall behind for large sizes. While NOG models remain closer to the target than ONE in all other cases, the results likewise show that the differentiable WIG setup further improves the performance. These networks reliably produce the best inference accuracy. Due to the stochastic nature of the non-linear learning processes, outliers exist, such as the 0.5m NOG model of the KOLM system. Nonetheless, the WIG models consistently perform best and yield an average improvement of 38% over the ONE baseline. Note that this error average behavior matches best-performing networks' behavior (Appendix D).

To conclude, our results allow for disentangling the influence of data shift and gradient divergence. As all training modalities, from data sets to random seeds, were kept constant, the only difference between NOG and WIG is the full gradient information provided to the latter. As such, we can deduce from our measurements that reducing the data shift contributes to the aforementioned improvement of 23%, while long-term gradient information yields another improvement of 15% (recommendation III). Figure 5 highlights this by depicting the accuracy of the unrolled setups normalized by the respective ONE setup for different model sizes and physical systems. WIG yields the best $L_2$ reductions. NOG in certain cases even performs worse with factors larger than one, due to its mismatch between loss landscape and gradient information, but nonetheless outperforms ONE on average.

**Unrolling horizons:** The effects of unrolling on data shift and gradient divergence depend on the unrolled length $m$, as summarized in figure 1. A long horizon diminishes the data shift. At the same time, gradient inaccuracy impairs the learning signal in the NOG case, while exploding gradients can

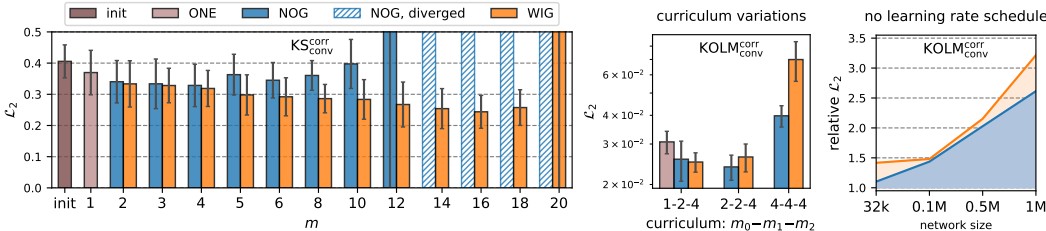

Figure 6: Left: variation of unrolling horizons $m$ trained for the KS system; all models were initialized with the leftmost ONE setup; with NOG training, the error increases and explodes for $m > 6$; WIG remains stable for longer horizons, diverges at $m > 18$; Middle: different curriculums for the KOLM system; starting the training with large $m$ leads to unfavorable network states; Right: Inference accuracy without learning rate schedule relative to with scheduling, i.e. $\mathcal{L}_2^{\text{no-schedule}}/\mathcal{L}_2^{\text{schedule}}$; All models benefit from learning rate scheduling, indicated by values are larger than one, and especially for larger models the schedule is crucial

have a similar effect for long backpropagation chains in the WIG setup. We test this hypothesis by varying the unrolled training horizon. The results are visualized in figure 6. For increasing horizons, the inference performance of NOG variants improves until a turning point is reached. After that, NOG performance deteriorates quickly. In the following regime, the stopped gradients cannot map the information gained by further reducing the data shift to an effective parameter update. The quality of the learning signal deteriorates. This is mitigated by allowing gradient backpropagation through the unrolled chain in the WIG setup. Herein, the inference accuracy benefits from even longer unrollings, and only diverges for substantially larger $m$ when instabilities from recurrent evaluations start to distort the direction of learning updates. These empirical observations confirm the theoretical analysis from Appendix B that went into Figure 1: There exist unrolling horizons for which training performance is improved for both NOG and WIG, while this effective horizon is longer for WIG.

As training with long unrolling poses challenges, we found a curriculum-based approach with an incremental increase of the unrolling length $m$ at training time to be essential (Um et al., 2020; Lam et al., 2022). Figure 6 shows how long unrollings in the initial training phases can hurt accuracy. At the same time, learning rate scheduling is necessary to stabilize gradients (recommendation IV).

**Gradient Stopping:** Cutting long chains of gradients was previously proposed as a remedy for training instabilities of unrolling (List et al., 2022; Brandstetter et al., 2022; Suh et al., 2022). Stopping gradients for a number of initial steps (Brandstetter et al., 2022; Prantl et al., 2022) is indicated by the parameter $w$ describing the number of *warm-up* steps for which gradients are discarded.

As shown in figure 7, setting $w$=1 can yield mild improvements over training over the full chain if no curriculum is used. Dividing the backpropagation into subsections (List et al., 2022) likewise does not yield real improvements in our evaluation. We divided the gradient chain into two or three subsections, identified by the parameter $v$ in figure 7. The best performance is obtained with the full WIG setup and curriculum learning, where unrolled models are pre-trained with $m$=1 models. This can be attributed

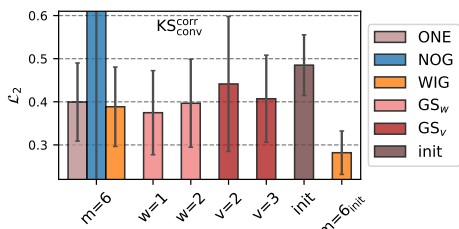

Figure 7: Gradient stopping techniques

to the more accurate gradients of WIG, as all gradient-stopping variants above inevitably yield a mismatch between loss landscape and learning updates (recommendation III).

**Size:** Figure 4 shows clear, continuous improvements in accuracy for increasing network sizes. This effect dominates the absolute error metrics. In line with Liu et al. (2022), we do not observe any "overfitting" effects even for models with millions of parameters applied to low-dimensional tasks like the KS system. Figure 8 estimates the convergence rate of the correction networks with respect to the parameter count to be $n^{-1/3}$. This convergence rate is poor compared to classic numerical solvers, albeit better rates could potentially be achieved by tailoring network architectures to specific problems. Nonetheless, a direct trade-off between numerical solvers and learned models can often be made for correction tasks. Thus, the benefits of NOG and WIG training are especially important

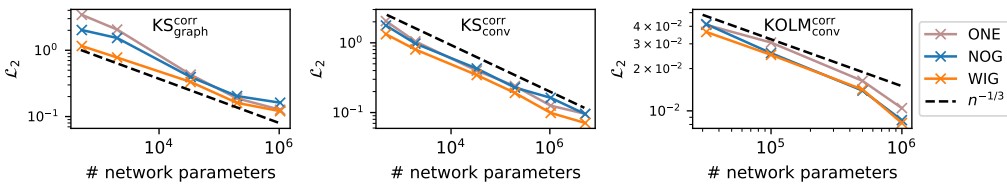

Figure 8: Accuracy convergence over network size, correction networks converge with $n^{-1/3}$

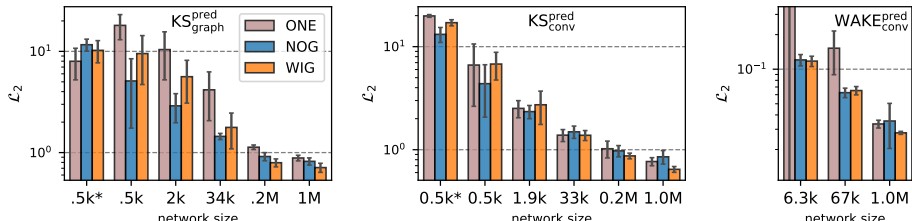

Figure 9: Prediction setups with inference accuracy measured in terms of $\mathcal{L}_2$ on KS and WAKE; NOG performs well for small network sizes, while WIG is show advantages for larger ones

for small- to medium-sized models that are relevant for most real-world applications where machine learning competes with purely numerical approaches (recommendation V). Hybrid approaches could benefit from the scaling of numerical solvers and the intrinsic benefits of learned models.

## 5.2 VARIED LEARNING TASKS

To broaden the investigation of the unrolling variants, we vary the learning task by removing the numerical solver from training and inference. This yields *prediction* tasks where the networks directly infer the desired solutions. Apart from this increased difficulty, all other training modalities were kept constant, i.e., we likewise compare non-differentiable NOG models to full unrolling (WIG).

**Prediction:** The parameter count of models still dominates the accuracy but in contrast to before, the NOG setup performs better than both alternatives for smaller network sizes. This is shown in figure 9. The inference errors for the prediction setup are roughly an order of magnitude larger than the respective correction errors. This indicates that pure predictions more quickly diverge from the reference trajectory, especially for small net-

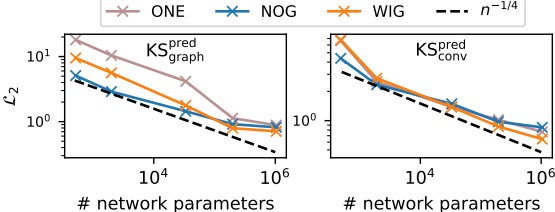

Figure 10: Accuracy convergence over network size, prediction converges with $n^{-1/4}$

work sizes. The deteriorated performance of WIG can be explained by these larger differences between the inferred state and the references, leading to suboptimal gradient directions. For the setups in Figure 9, NOG training achieves improvements of 31%, and WIG improves on this by a further 3.6%. In general, unrolling maintains strong benefits in prediction setups, but differentiability is less beneficial than in correction setups.

**Smooth Transition:** While the learning task is typically mandated by the application, our setup allows us to investigate the effects of unrolling for a smooth transition from prediction to increasingly simple correction tasks, in line with figure 2. In the pure prediction case, the physical prior does not model any physics, i.e., is an identity operator. We transition away from pure predictions by providing the network with improving inputs by increasing the time step of the reference solver for correction tasks. On the other end of the spectrum the numerical solver computes the full time step, and the neural network now only has the trivial task to provide an identity function. Since the total error naturally decreases with simpler tasks, figure 11 shows the performance normalized by the task difficulty. The positive effects of NOG and especially WIG training carry over across the full range of tasks. Interestingly, the NOG version performs best for very simple tasks on the right sides of each graph. This is most likely caused by a relatively small mismatch of gradients and energy landscape. In our experiments, the task difficulty was changed by artificially

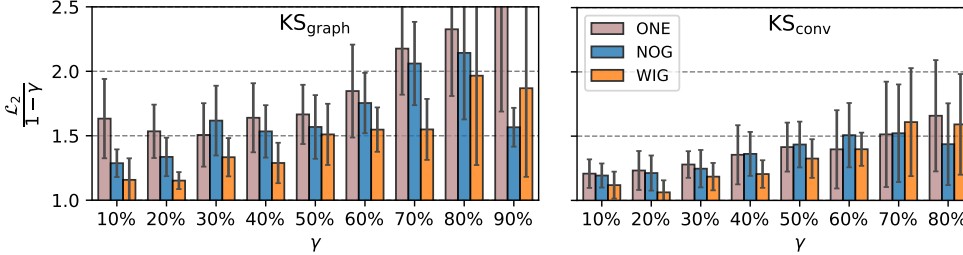

Figure 11: Transition from prediction to correction; a smooth transition in modeling difficulty is achieved by varying the timestep of the numerical solver embedded in the correction step $\gamma = \Delta t_{\mathcal{S}}/\Delta \tilde{t}$, with the solver timestep $\Delta t_{\mathcal{S}}$ and the ground truth timestep $\Delta \tilde{t}$

varying the prior's accuracy. This mimics the effects of basing the correction setups on different numerical schemes. Since unrolling manifests a stable performance improvement across all priors, it promises benefits for many correction setups in other applications. The results above indicate that a 5x accuracy boost can be achieved by integrating low-fidelity physics priors in a NOG setup.

**Changing the Physical System:** The AERO dataset comprises aerofoil flows in the transonic regime and features shocks, fast-changing vortical structures, and diverse samples around the critical Ma = 0.8. As such, we deployed a modern attention-based U-Net (Oktay et al., 2018). As a complex, large-scale test scenario, our learning setups for the AERO system implement the recommendations we derived from the previous results. Figure 12 shows the inference evaluation of our trained models. Once again, unrolling increases accuracy at inference time. On average, unrolling improves the infer-

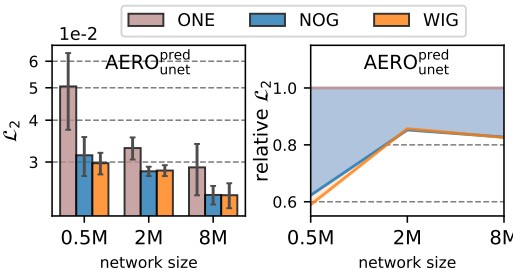

Figure 12: Left: Prediction of the AERO system, absolute errors of ONE, NOG, and WIG; Right: Relative errors with respect to ONE

ence loss by up to 32% in the AERO case. The results reflect our recommendations: Network size has the largest impact, and choosing the right size balances performance and accuracy (V). Additionally, our models were trained with a learning rate scheduled curriculum to achieve the best results (IV), details of which are found in Appendix C.4. Training on the AERO system resembles the behavior of lower-dimensional problems, e.g. the KS system (I). Similar to our previous prediction tasks, long-term gradients are less crucial, but still deliver the best models (III). However, unrolling itself is essential for stable networks (II).

## 6    CONCLUSION

We have conducted in-depth empirical investigations of unrolling strategies for training neural PDE simulators. The inherent properties of unrolling were deduced from an extensive test suite spanning multiple physical systems, learning setups, network architectures, and network sizes. Our findings rendered five best practices for training autoregressive neural simulators via unrolling. Additionally, our test sets and differentiable solvers are meant to serve as a benchmark: The broad range of network sizes for popular architectures as well as the selection of common physical systems as test cases yield a flexible baseline for future experiments with correction and prediction tasks.

Nevertheless, there are limitations to our scope. Our physical systems live in the domain of non-linear chaos and are primarily connected to fluid mechanics. As a consequence, there is naturally no guarantee that our results will directly carry over to other domains. Additionally, we mentioned the relevance of our results to the scientific computing community. Our recommendations can help prioritize implementation efforts when designing unrolled training setups. At the same time, only our AERO system matches the usual complexity in scientific computing. Lastly, we have only studied a subset of the relevant hyperparameters. In the context of unrolling, variations such as network width and depth, advanced gradient-stopping techniques, and irregularly spaced curriculums could be impactful. These topics are promising avenues for future research to further our understanding of autoregressive neural networks for scientific applications.

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

# APPENDIX

## A  GRADIENT CALCULATION

### A.1  CORRECTION

The network parameter optimization was defined in Equation 1. For one learning iteration, a loss was accumulated over an unrolled trajectory. We can write this total loss over the unrolled trajectory as a function of $g(\mathbf{u}^i) = f_\theta(\mathcal{S}(\mathbf{u}^i))$ such that

$$\mathcal{L} = \sum_{i=1}^{s} \mathcal{L}_2(\tilde{\mathbf{u}}^{i+\tau s}, g^s(\mathbf{u}^i)) = \sum_{i=1}^{s} \mathcal{L}^s, \tag{2}$$

where $g^s$ represents the recurrent application of multiple machine learning augmented simulation steps, and $\mathcal{L}^s$ represents the loss evaluated after that step. Note that the full backpropagation through this unrolled chain requires a differentiable solver for the correction setup. To test the effect of using a differentiable solver, we introduce two different strategies for propagating the gradients $\frac{\partial \mathcal{L}_u}{\partial \theta}$. The differentiable setup can calculate the full optimization gradients by propagating gradients through the solver. The gradients are thus evaluated as

$$\frac{\partial \mathcal{L}^s}{\partial \theta} = \sum_{B=1}^{s} \left[ \frac{\partial \mathcal{L}^s}{\partial g^s} \left( \prod_{b=s}^{B+1} \frac{\partial g^b}{\partial f_\theta^b} \frac{\partial f_\theta^b}{\partial \mathcal{S}^b} \frac{\partial \mathcal{S}^b}{\partial g^{b-1}} \right) \frac{\partial g^B}{\partial f_\theta^B} \frac{\partial f_\theta^B}{\partial \theta} \right]. \tag{3}$$

We refer to this fully differentiable setup as WIG. In contrast, if no differentiable solver is available, optimization gradients can only propagate to the network application, not through the solver. The gradients are thus evaluated as

$$\frac{\partial \mathcal{L}_s}{\partial \theta} = \frac{\partial \mathcal{L}_s}{\partial g^s} \frac{\partial g^s}{\partial f_\theta^s} \frac{\partial f_\theta^s}{\partial \theta}. \tag{4}$$

This setup is referred to as NOG. Most existing code bases in engineering and science are not fully differentiable. Consequentially, this NOG setup is particularly relevance, as it could be implemented using existing traditional numerical solvers.

### A.2  PREDICTION

Prediction operates on the same network parameter optimization from Equation 1. The total loss over the unrolled trajectory is

$$\mathcal{L} = \sum_{i=1}^{s} \mathcal{L}_2(\tilde{\mathbf{u}}^{i+\tau s}, f_\theta^s(\mathbf{u}^i)) = \sum_{i=1}^{s} \mathcal{L}^s, \tag{5}$$

where $f_\theta^s$ represents the recurrent application of the network and $\mathcal{L}^s$ represents the loss evaluated after that step. To test the effect of using long-term gradients, we introduce two different strategies for propagating the gradients $\frac{\partial \mathcal{L}_u}{\partial \theta}$. The differentiable setup can calculate the full optimization gradients by propagating gradients through the solver. The gradients are thus evaluated as

$$\frac{\partial \mathcal{L}^s}{\partial \theta} = \sum_{B=1}^{s} \left[ \frac{\partial \mathcal{L}^s}{\partial f_\theta^s} \left( \prod_{b=s}^{B+1} \frac{\partial f_\theta^b}{\partial f_\theta^{b-1}} \right) \frac{\partial f_\theta^B}{\partial \theta} \right]. \tag{6}$$

We refer to this fully differentiable setup as WIG. In contrast, if no differentiable solver is available, optimization gradients can only propagate to the network application, not through the solver. The gradients are thus evaluated as

$$\frac{\partial \mathcal{L}_s}{\partial \theta} = \frac{\partial \mathcal{L}_s}{\partial f_\theta^s} \frac{\partial f_\theta^s}{\partial \theta}. \tag{7}$$

This setup is referred to as NOG. Most existing code bases in engineering and science are not fully differentiable. Consequentially, this NOG setup is particularly relevance, as it could be implemented using existing traditional numerical solvers.

# B  UNROLLING CHAOTIC SYSTEMS

The following sections comprise a theoretical analysis of learning chaotic dynamics. For ease of notation, we conduct this analysis for predictive setups. Such learning tasks focus on training a mapping $f_\theta$ that reproduces the dynamics of a ground truth $f$, without any numerical solver in the loop. These considerations can be trivially expanded to correction setups.

## B.1  DATA SHIFT

Suppose we study a chaotic dynamical system $\tilde{u}_{n+1} = f(\tilde{u}_n, \nabla \tilde{u}, \nabla^2 \tilde{u}, ...)$, where $\tilde{u}$ represents a discrete ground-truth state. The chaotic dynamics drive this system to an attractor $A_f$ representing a subset of the phase space of $\tilde{u}$. In prediction and correction tasks, a neural simulator learns the dynamics

$$u_{n+1} = f_\theta(u_n, \nabla u, \nabla^2 u...), \tag{8}$$

which approximates the evolution of the ground truth system. Since perfect reproduction of $f$ is generally not achieved, differences between the trajectories of $\tilde{u}$ and $u$ exist, leading to $A_f \neq A_{f_\theta}$. Crucially, there is no guarantee for ONE training that the states $u$ observed during training sufficiently represent $A_{f_\theta}$. In other words,

$$\forall \tilde{u}_i \in A_f, A_{f_\theta} : \lim_{N \to \infty} \{f_\theta(\tilde{u}_0), f_\theta(\tilde{u}_1), ..., f_\theta(\tilde{u}_N)\} \neq A_{f_\theta}. \tag{9}$$

This means that we are not guaranteed to explore the attractor of the learned system $A_{f_\theta}$ when only observing states based on one discrete evolution of $f_\theta$, regardless of the dataset size $N$.

Let us now suppose we unroll $m$ steps during training such that $u^m = f_\theta^m(u)$, where $f_\theta^m$ denotes the autoregressive evolution of $m$ steps. Based on the definition of the attractor $A_{f_\theta}$ we can state that

$$\lim_{m \to \infty} \{f_\theta^0(\tilde{u}), f_\theta^1(\tilde{u}), ..., f_\theta^m(\tilde{u})\} = A_{f_\theta}. \tag{10}$$

In contrast to ONE training, unrolled training thus exposes the inference attractor $A_{f_\theta}$ at training time for sufficiently large $m$. Precisely the difference between the observed training set (sampled from $A_f$) and the inference attractor $A_{f_\theta}$ is commonly known as data shift in machine learning. The dark red curve in Figure 1 shows how this data shift is thus reduced by choosing larger $m$.

## B.2  GRADIENTS IN UNROLLED SYSTEMS

In Table 1 and Appendix A we derived the precise gradient equations for NOG and WIG setups. These are

$$\frac{\partial \mathcal{L}_{NOG}}{\partial \theta} = \sum_{s=1}^{m} \frac{\partial \mathcal{L}^s}{\partial f_\theta^s} \frac{\partial f_\theta^s}{\partial \theta}, \qquad \frac{\partial \mathcal{L}_{WIG}}{\partial \theta} = \sum_{s=1}^{m} \sum_{B=1}^{s} \left[ \frac{\partial \mathcal{L}^b}{\partial f_\theta^b} \left( \prod_{b=s}^{B+1} \frac{\partial f_\theta^b}{\partial f_\theta^{b-1}} \right) \frac{\partial f_\theta^B}{\partial \theta} \right]. \tag{11}$$

Note that WIG unrolling calculates the true gradient. We can thus derive the gradient inaccuracy of the NOG setup as

$$\frac{\partial \mathcal{L}_{WIG}}{\partial \theta} - \frac{\partial \mathcal{L}_{NOG}}{\partial \theta} = \sum_{s=1}^{m} \sum_{B=1}^{s-1} \left[ \frac{\partial \mathcal{L}^b}{\partial f_\theta^b} \left( \prod_{b=s-1}^{B+1} \frac{\partial f_\theta^b}{\partial f_\theta^{b-1}} \right) \frac{\partial f_\theta^B}{\partial \theta} \right] \tag{12}$$

We can now integrate a property of chaotic dynamics derived by Mikhaeil et al. (2022). Herein, the authors show that for chaotic systems, the Jacobians $\mathbf{J_s} = \frac{\partial f_\theta^s}{\partial f_\theta^{s-1}}$ have eigenvalues larger than 1 in the geometric mean. Thus,

$$\left\| \prod_{b=s-1}^{B+1} \frac{\partial f_\theta^b}{\partial f_\theta^{b-1}} \right\| > 1 \tag{13}$$

holds in this case. This means that the gradient inaccuracy increases with the unrolling. We can also observe that the NOG gradient inaccuracy grows with $\propto m^2$, while the NOG gradients only linearly depend on $m$. As a consequence, the gradients computed in the NOG system diverge from the true (WIG) gradients for increasing $m$, as shown with the blue line in Figure 1. The gradient

approximations used in NOG thus do not accurately match the loss used in training. This hinders the network optimization.

Let us finally consider the full gradients of the WIG setup themselves. We can use Theorem 2 from Mikhaeil et al. (2022), which states that

$$\lim_{m \to \infty} ||\frac{\partial f_\theta^m}{\partial f_\theta^0}|| = \infty \tag{14}$$

for almost all points on $A_{f_\theta}$. The gradients of the WIG setup explode exponentially for long horizons $m$, but less quickly than those of NOG, as indicated by the orange line below the blue one in Figure 1. As a direct consequence, the training of the WIG setup becomes unstable for chaotic systems when $m$ grows to infinity.

We can summarize the above findings as follows. Unrolling training trajectories for data-driven learning of chaotic systems reduces the data-shift, as the observed training samples converge to the learned attractor. At the same time, long unrollings lead to unfavorable gradients for both NOG and WIG setups. A range of medium-sized unrolling horizons might exist, where the benefits of reducing the data-shift outweigh instabilities in the gradient. These setups are studied in the paper and exposed in figure 6 for the KS system.

## C    DETAILS OF PHYSICAL SYSTEMS AND ARCHITECTURES

### C.1    KURAMOTO-SIVASHINSKY CASE (KS)
**Numerical data:**    The KS equation is a fourth-order stiff PDE governed by

$$\frac{\partial u}{\partial t} + u\frac{\partial u}{\partial x} + \frac{\partial^2 u}{\partial x^2} + \frac{\partial^4 u}{\partial x^4} = 0, \tag{15}$$

with simulation state $u$, time $t$, and space $x$ living in the domain of size $\mathcal{X}$. The fourth-order term leads to a highly chaotic behavior. The equation was simulated using a second-order exponential time-stepping solver following the ETRK2 scheme in (Cox & Matthews, 2002). The domain was discretized with 48 grid-points, and timesteps were set to 1. The physical domain length $\mathcal{X}$ is the critical parameter in the KS equation. The training dataset was computed for a range of $\mathcal{X} = [5.6, 6.4, 7.2, 8, 8.8, 9.6, 10.4]$. A sequence of 5000 steps was computed for each domain length. The numerical solves and network training were performed in PyTorch (Paszke et al., 2019).

**Neural Network Architectures:**    We use two architectures for learning tasks with the KS case: a graph-based (GCN) and a convolutional network (CNN). Both networks follow a ResNet-like structure (He et al., 2016), have an additive skip connection from input to output, and are parameterized in terms of their number of *features* per message-passing step or convolutional layer, respectively. They receive 2 channels as input, a normalized domain size $\mathcal{X}$ and $u$, and produce a single channel, the updated $u$, as output. The GCN follows the hierarchical structure of directional Edge-Conv graph nets Wang et al. (2019); Li et al. (2019) where each Edge-Conv block is comprised of two message-passing layers with an additive skip connection. The message-passing concatenates node, edge, and direction features, which are fed to an MLP that returns output node features. At a high level, the GCN can be seen as the message-passing equivalent of a ResNet. These message passing layers in the GCN and convolutional layers in the CNN are scaled in terms of whole ResBlocks, i.e., two layers with a skip connection and leaky ReLU activation. In the following we denote the number of blocks as network *depth*, and specify the network architectures with a tuple containing (features, depth). The networks feature additional linear encoding and decoding layers for input and output. The architectures for GCN and CNN were chosen such that the parameter count matches, as listed in table 2.

For both GCN and CNN the smallest network size with a depth indicated by "-" is a special case using a single non-linear layer with the listed number of features. Hence, this represents the smallest possible architecture for our chosen range of architectures, consisting of a linear encoding, one non-linear layer followed by a linear decoding layer.

**Network Training:**    Every network was trained for a total of 50000 iterations of batchsize 16. The learning rate was initially set to $1 \times 10^{-4}$, and a learning rate decay of factor 0.9 was deployed after every 2000 iterations during training. The KS systems was set up to quickly provide substantial

Table 2: KS Network Architectures and Parameter Counts

| *Architecture*: | .5k* | .5k | 2k | 33k | .2M | 1M |
|---|---|---|---|---|---|---|
| GCN (features, depth): | (46,-) | (9,1) | (14,2) | (31,8) | (63,12) | (126,16) |
| GCN Parameters: | 511 | 518 | 2007 | 33,578 | 198,770 | 1,037,615 |
| CNN (features, depth): | (48,-) | (8,1) | (12,2) | (26,8) | (52,12) | (104,16) |
| CNN Parameters: | 481 | 481 | 1897 | 33125 | 196457 | 1042705 |

changes in terms of simulated states. As a consequence, short unrollments of $m = 3$ were sufficient during training unless otherwise mentioned. For this unroll number, no curriculum was necessary.

For fairness across the different training variants the KS setup additionally keeps the number of training data samples constant for ONE versus NOG and WIG. For the latter two, unrolling means that more than one state over time is used for computing the learning gradient. Effectively, these two methods see $m$ times more samples than ONE for each training iteration. Hence, we increased the batch size of ONE training by a factor of $m$. However, we found that this modification does not result in an improved performance for ONE in practice.

For the tests shown in figure 6 of the main paper we varied the training data set size, reducing the number of time steps in the training data set to 1000, while keeping the number of training iterations constant. The largely unchanged overall performance indicates that the original data set could be reduced in size without impeding the performance of the trained models.

**Network Evaluation:** We computed extra- and interpolative test sets for the domain length $\mathcal{X}$. The extrapolation uses $\mathcal{X} = [4.8, 11.2]$, whilst interpolation is done for $\mathcal{X} = [6.8, 9.2]$.

For our $\mathcal{L}_2$ results, we initialized autoregressive runs with 20 initial conditions for each $\mathcal{X}$, accumulated the $\mathcal{L}_2$ for sequences of 40 steps, and averaged over all extra- and interpolative cases. The standard deviation is computed over the total set of tests. The time until decorrelation was calculated by running an autoregressive inference sequence. The steps were counted until the cross-correlation between the inferred state and the reference dropped below 0.8. The statistics of the reached step-counts were gathered over all test cases, just like for the $\mathcal{L}_2$. The divergence time was defined as the number of inference steps until $\mathcal{L}_2 > 500$. Again, statistics were gathered over all test cases.

## C.2 UNSTEADY WAKE-FLOW CASE (WAKE)

**Numerical data:** The WAKE case uses numerical solutions of the incompressible Navier-Stokes equations

$$\frac{\partial \mathbf{u}}{\partial t} + (\mathbf{u} \cdot \nabla)\mathbf{u} = -\nabla p + \nu \nabla^2 \mathbf{u},$$
$$\nabla \cdot \mathbf{u} = 0, \tag{16}$$

with the two-dimensional velocity field $\mathbf{u}$ and pressure $p$. Simulations were run for a rectangular domain with a 2:1 side aspect ratio, and a cylindrical obstacle with a diameter of $L_D = 0.1$ placed at $(0.5, 0.5)$ with a constant bulk inflow velocity of $U = 1$. The Reynolds number is defined as $Re = \frac{U L_D}{\nu}$. The ground truth data was simulated with operator splitting using a Chorin-projection for pressure and second-order semi-Lagrangian advection Bridson (2015). The physical behavior is varied with the Reynolds number by changing the viscosity of the fluid. The training data set contains 300 time steps for six Reynolds numbers $Re = [97.5, 195, 390, 781.25, 1562.5, 3125]$ at resolution $256, 128$. For learning tasks, these solutions are down-sampled by $4\times$ to $64, 32$. The solver for the correction uses the same solver with a more diffusive first-order advection step. The numerical solves and network training were performed in PyTorch (Paszke et al., 2019).

**Neural Network Architectures:** This test case employs fully-convolutional residual networks (He et al., 2016). The architectures largely follow the KS setup: the neural networks have a ResNet structure, contain an additive skip connection for the output, and use a certain number of ResBlocks with a fixed number of features. The *smallest* network uses 1 ResBlock with 10 features, resulting in 6282 trainable parameters. The *medium*-sized network has 10 ResBlocks with 16 features and

66,178 parameters, while the *large* network has 20 blocks with 45 features resulting in more than 1 milllion (1,019,072) parameters.

**Network Training and Evaluation:**   The networks were trained for three steps with 30000 iterations each, using a batch size of 3 and a learning rate of $1 \times 10^{-4}$. The training curriculum increases the number of unrolled steps (parameter $m$ of the main text) from $m = 1$, to $m = 4$ and then $m = 16$. Each stage applies learning rate decay with a factor of $1/10$.

$\mathcal{L}_2$ errors are computed and accumulated over 110 steps of simulation for 12 test cases with previously unseen Reynolds numbers: three interpolative ones $Re = [2868.5, 2930, 2990]$ and nine extrapolative Reynolds numbers $Re = [3235, 3174, 3296, 3845, 3906, 3967, 4516.5, 4577.5]$.

### C.3   KOLMOGOROV TURBULENCE (KOLM)

**Numerical data:**   The two-dimensional Kolmogorov turbulence is governed by the incompressible Navier-Stokes equations with an additive forcing term

$$\frac{\partial \mathbf{u}}{\partial t} + (\mathbf{u} \cdot \nabla)\mathbf{u} = -\nabla p + \frac{1}{Re}\nabla^2 \mathbf{u} + \mathbf{f}, \tag{17}$$
$$\nabla \cdot \mathbf{u} = 0,$$

where $\mathbf{u}$ and $p$ are velocity and pressure fields respectively. The additive forcing causes the formation of a shear layer, whose instability onset develops into turbulence (Givental et al., 2009). The forcing was set to $\mathbf{f} = [\sin(k_x * y),\ 0]^T$ with wavenumber $k_x = 6$. The simulation used a second-order semi-implicit PISO scheme (Issa, 1986). The physical domain size was set to $L_x = L_y = 2\pi$ and discretized by a $128 \times 128$ grid. The timestep was set to $\Delta t = 0.005$ for the high-resolution ground truth dataset, which maintained a Courant number smaller than $0.5$. The training dataset is based on a Reynolds number variation of $Re = [300, 400, 500, 700, 800, 900]$. For each Reynolds number, 6000 frames were added to the dataset. The numerical solves and network training were performed in Tensorflow (Abadi, 2016).

**Neural Network Architectures:**   The KOLM networks are also based on ResNets. In each ResNet block, data is processed through convolutions and added to a skip connection. In contrast to KS and WAKE architectures, the number of features varies throughout the network. Four different network sizes were deployed, details of which are listed in Table 3. The network sizes in the KOLM case range from 32 thousand to 1 million.

Table 3: KOLM Network Architectures and Parameter Counts

| Architecture | # Parameters | # ResNet Blocks | Block-Features |
|---|---|---|---|
| CNN, 32k | 32369 | 5 | [8, 20, 30, 20, 8] |
| CNN, 0.1M | 115803 | 7 | [8, 16, 32, 64, 32, 16, 8] |
| CNN, 0.5M | 461235 | 7 | [16, 32, 64, 128, 64, 32, 16] |
| CNN, 1M | 1084595 | 9 | [16, 32, 64, 128, 128, 128, 64, 32, 16] |

**Network Training and Evaluation:**   A spatiotemporal downsampling of the ground truth data formed the basis of the training trajectories. Thus, training operated on a $4\times$ downsampled resolution with 4 times larger time-steps, i.e. $32 \times 32$ grid with $\Delta t = 0.02$. The networks were trained with a curriculum that incrementally increased the number of unrolled steps such that $m = [1, 2, 4]$, with an accompanying learning rate schedule of $[10^{-4}, 10^{-5}, 10^{-6}]$. Additionally, a learning rate decay with factor $0.9$ after an epoch of 36 thousand iterations. For each $m$, 144 thousand iterations were performed. The batch size was set to $1$.

$\mathcal{L}_2$ errors are computed and accumulated over 250 steps of simulation for 5 test cases with previously unseen Reynolds numbers: $Re = 600$ for interpolation and $Re = 1000$ for extrapolation.

### C.4   AERO CASE

**Numerical data:**   The governing equations for the transonic flow over a NACA0012 airfoil are non-dimensionalized with the freestream variables (i.e., the density $\rho_\infty$, speed of sound $a_\infty$, and the

chord length of the airfoil $c$), and can be expressed in tensor notation as

$$\frac{\partial \rho u_i}{\partial x_i} = 0$$

$$\frac{\partial \rho u_i u_j}{\partial x_j} = -\frac{\partial p}{\partial x_i} + \frac{\partial \tau_{x_i x_j}}{\partial x_j} \qquad (18)$$

$$\frac{\partial (\rho E + p) u_i}{\partial x_i} = \frac{\partial (-q_i + u_j \tau_{x_i x_j})}{\partial x_i}$$

where the shear stress (with Stokes' hypothesis) and heat flux terms are defined as

$$\tau_{x_i x_j} = \mu \frac{M_\infty}{Re_\infty} [(\frac{\partial u_i}{\partial x_j} + \frac{\partial u_j}{\partial x_i}) - \frac{2}{3} \frac{\partial u_k}{\partial x_k} \delta_{ij}]$$

and

$$q_{x_i} = -\frac{\mu}{Pr} \frac{M_\infty}{Re_\infty (\gamma - 1)} \frac{\partial \Theta}{\partial x_i}.$$

Here, Reynolds number is defined as $Re_\infty = \rho_\infty \sqrt{u_\infty^2 + v_\infty^2} c / \mu_\infty$; $\gamma$ is the ratio of specific heats, 1.4 for air; the laminar viscosity $\mu$ is obtained by Sutherland's law (the function of temperature), and the turbulent viscosity $\mu_T$ is determined by turbulence models; laminar Prandtl number is constant, i.e. $Pr = 0.72$. The relation between pressure $p$ and total energy $E$ is given by

$$p = (\gamma - 1)[\rho E - \frac{1}{2} \rho u_i u_i].$$

Note also that from the equation of state for a perfect gas, we have $p = \rho a^2 / \gamma$ and temperature $\Theta = a^2$. As we perform 2D high-resolution quasi-direct numerical simulations, no turbulence model is employed.

The finite-volume method numerically solves the equations using the open-source code CFL3D. The mesh resolution $1024 \times 256$ is kept the same for cases, i.e. 256 grid cells in the wall-normal direction, 320 grid cells in the wake, and 384 grid cells around the airfoil surface. The convective terms are discretized with third-order upwind scheme, and viscous terms with a second-order central difference.

An inflow/outflow boundary based on one-dimensional Riemann invariant is imposed at about $50c$ away from the airfoil in the $(x, y)$ plane. The grid stretching is employed to provide higher resolution near the surface and in the wake region, and the minimal wall-normal grid spacing is $6 \times 10^{-4}$ to ensure $y_n^+ < 1.0$. A no-slip adiabatic wall boundary condition is applied on the airfoil surface. The non-dimensional time step is $0.008c/U_\infty$.

In the transonic regime, airflow behavior becomes more complex due to the formation of shock waves and supersonic and subsonic flow areas on the airfoil surfaces. When the Mach number is below $M = 0.77$, the unsteadiness in the flowfield is mainly caused by the high-frequency vortex shedding. At around $M = 0.8$, a series of compression waves coalesce to form a strong shock wave, and the flow structures are dominated by alternately moving shock waves along the upper and lower sides of the airfoil. For $M > 0.88$, the strong shock waves become stationary on both surfaces. To cover all possible flow regimes, the samples in the training dataset are generated at $M = [0.75, 0.8, 0.825, 0.88, 0.9]$, and the test samples are performed at $M = 0.725$ and $M = 0.775$.

The snapshots are saved at every four simulation steps (i.e., $dt_{sampling} = 0.032c/U_\infty$) and spatially downsampled by 4x and 2x in the circumferential and wall-normal directions, respectively. In the case of $M = 0.85$, there are 1000 snapshots; for other cases, there are 500 snapshots.

**Neural Network Architectures:** We implemented Attention U-Net (Oktay et al., 2018). It consists of three encoder blocks, each progressively capturing features from the input image through convolution and downsampling. Following the encoder blocks, there's a bottleneck block for information compression with a higher number of channels. Subsequently, the architecture includes three decoder blocks, which use skip connections to integrate features from both the bottleneck and corresponding encoder blocks during the upscaling process. These decoder blocks gradually reduce the number of channels, culminating in a 1x1 convolutional output layer that generates pixel-wise predictions. We train networks of varying sizes by adjusting the number of features, as indicated in Table 4. The training was performed in PyTorch (Paszke et al., 2019).

Table 4: AERO Network Architectures and Parameter Counts

| Architecture | # Parameters | Features in encoder & bottleneck blocks |
|---|---|---|
| UNet, 0.5m | 511332 | [16, 32, 64] [128] |
| UNet, 2.0m | 2037956 | [32, 64, 128] [256] |
| UNet, 8.1m | 8137092 | [64, 128, 256] [512] |

**Network Training and Evaluation:** The model is trained with Adam and a mini-batch size of 5, with training noise, for up to 500k iterations. A learning rate of $6 \times 10^{-4}$ is used for the first 250k iterations and then decays exponentially to $6 \times 10^{-5}$. The training curriculum increases the number of unrolled steps from $m = 1$ to $m = 4$ and then $m = 9$.

$\mathcal{L}_2$ errors are computed and accumulated over 200 prediction steps (equivalent to 800 simulator steps) of simulation for two test cases with previously unseen Mach numbers: $M = 0.725$ and $M = 0.775$, corresponding to shock-free case and near-critical condition case.

## D    ADDITIONAL RESULTS

### D.1    INTERPOLATION AND EXTRAPOLATION TESTS

Interpreting physical hyperparameters is necessary for models to generalize to extrapolative test cases. We want to test whether extrapolation to new physical hyperparameters benefits from unrolling or long term gradients. In the main sections, all evaluations accumulated their results from interpolative and extrapolative test cases. In this section, we explicitly differentiate between interpolation and extrapolation tests. Figures 13, 14, 15, and 16 compare the model performance on interpolative and extrapolative test cases with respect to the physical parameters and depicts performance relative to the ONE baseline for various model sizes. Overall accuracy is worse on the harder extrapolative test cases. Similarly to the combined tests from the main section WIG performs best on average, both on interpolative or extrapolative data. However, the long-term gradients introduced by this method do not seem to explicitly favor inter- or extrapolation. Nonetheless, the positive aspects of WIG training are not constrained to interpolation, but successfully carry over to extrapolation cases.

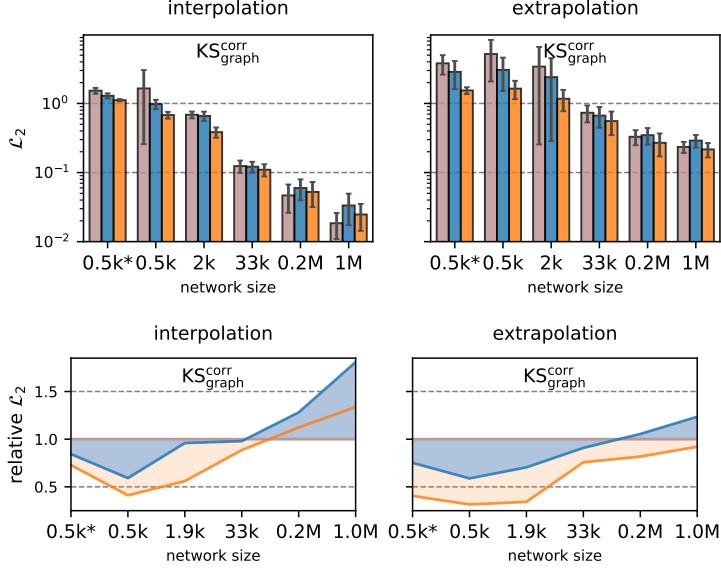

Figure 13: Comparison of $\mathcal{L}_2$ errors on interpolative and extrapolation test sets for GCNs on KS

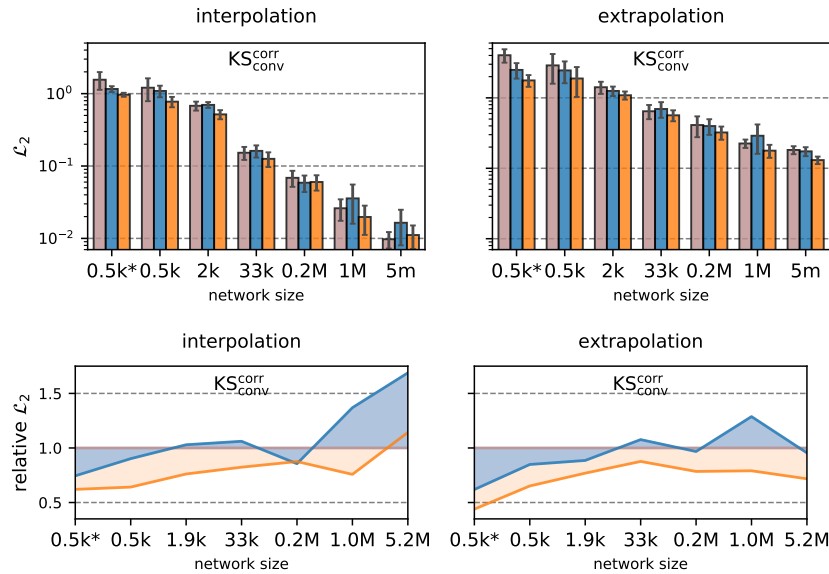

Figure 14: Comparison of $\mathcal{L}_2$ errors on interpolative and extrapolation test sets for CNNs on KS

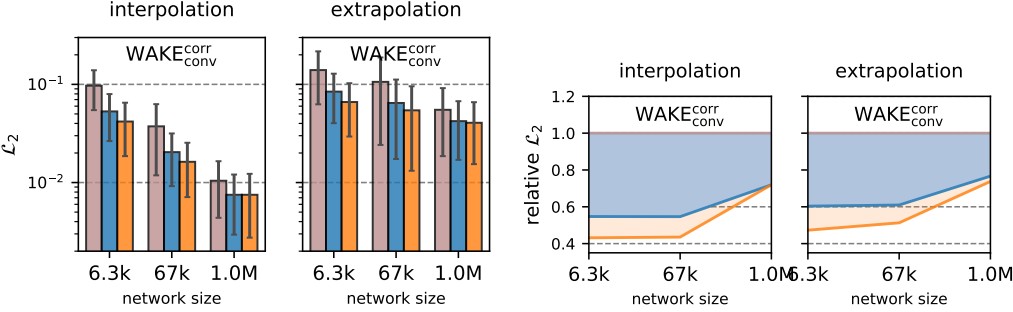

Figure 15: Comparison of $\mathcal{L}_2$ errors on interpolative and extrapolation test sets for WAKE

## D.2 DATASET SIZES

Unrolling connects multiple dataset frames in one trajectory. This connectedness encodes the physical relation between subsequent frames that ultimately has to be learned by the neural network. Given that unrolled setups observe more of these physical connections, one might expect that unrolling decreases the necessary dataset size. To test this hypothesis, we incrementally decreased the amount of training data. The number of training iterations was kept constant throughout the process. The models are then evaluated on our full test sets. Figure 17 compares the $\mathcal{L}_2$ for variations in the training dataset size. A measurable difference in inference accuracy only appears for dataset sizes smaller than $X\%$ of the original dataset. While the accuracy of WIG does indeed deteriorate slightly later than NOG and especially ONE, this transition is confined to a small section of $Y\%$ dataset size. In practice, training a setup in this narrow dataset margin is unlikely and thus not mentioned as a strong benefit of unrolling.

## D.3 BEST PERFORMING MODELS

Our error measurements relied on statistical evaluations, which in turn were based on multiple randomized training runs. NOG and especially WIG showed clear benefits in these statistical evaluations. However, we must also consider the computational cost of training (differentiable) unrolled setups. Unrolling $m$ steps with the NOG setup increases the computational cost of one training

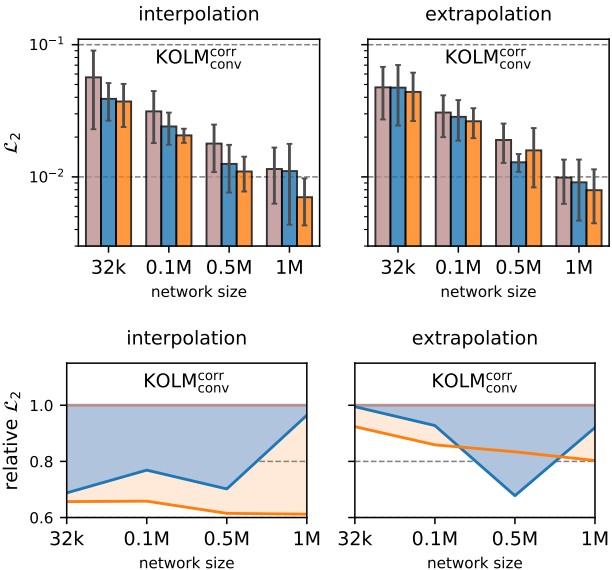

Figure 16: Comparison of $\mathcal{L}_2$ errors on interpolative and extrapolation test sets for KOLM

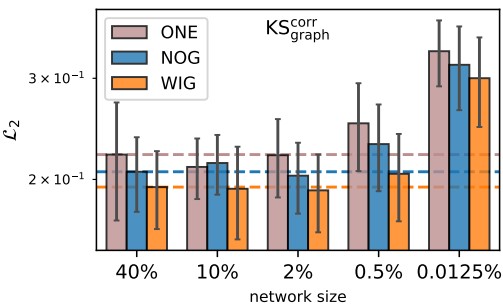

Figure 17: Dataset size variations in percent of the full training set

iteration $m$-fold. Differentiable training in the WIG setup adds even more computations due to the backpropagation through the solver. Consequently, training multiple models and selecting the best might be a viable approach for ONE models, where training costs are lower. Thus, we evaluate the best-performing models in a separate analysis. Figure 18 depicts the best inference $\mathcal{L}_2$ achieved by a given model size, architecture, and learning setup. The best ONE models are on par with NOG or WIG for some network sizes and architectures. However, in many other cases, the average unrolled WIG setup still performs better or similar to the best ONE model. In light of the fact that these best models were selected out of 8 (KOLM) or 20 (KS) training runs, training with unrolling is ultimately more resource-efficient if best performance is sought. Thus, our recommendation of training with NOG or WIG approaches persists.

## D.4 EVALUATION METRICS

Our main evaluations used the $\mathcal{L}_2$ to quantify network accuracy. Since the $\mathcal{L}_2$ was also used as a training loss, a good inference performance indicates a less pronounced data shift during inference and stable gradients during training. This level of interpretability is not given for other metrics, motivating the choice of $\mathcal{L}_2$ as the main test metric. Nevertheless, good performance on other metrics, such as correlation, is desirable for learned simulators. The overall conclusions from the main paper can similarly be deduced from these other metrics. Figure 19 depicts the time until de-correlation between inference runs and the ground truth for the KS system. For all model sizes and both network architectures, WIG achieves the longest inference rollouts until the de-correlation

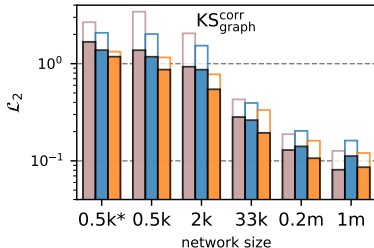 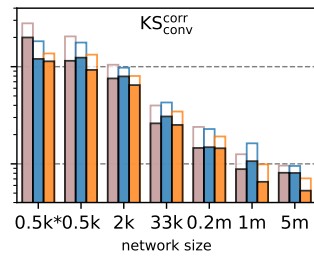 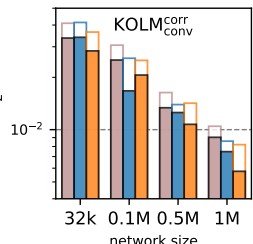

Figure 18: $\mathcal{L}_2$ errors of the best performing models; the outline in the background represent the average accuracy

threshold is reached. Similarly to the observations on the $\mathcal{L}_2$ metric in Figure 4, NOG is the second best option for small and medium-sized networks when it comes to correlation. This observation again translates between the network architectures.

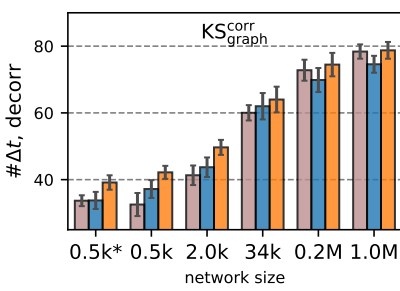 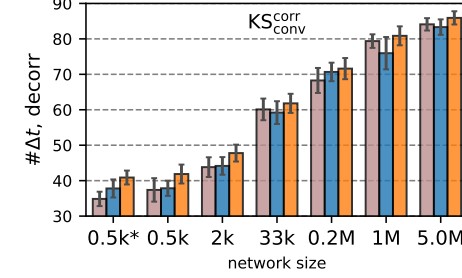

Figure 19: Number of timesteps until the threshold is reached for de-correlation

In addition, we studied the time until divergence for the KS system. This metric sets a high threshold of $\mathcal{L}_2 > 500$. At the same time, the solution state of a simulation ranges within $|\mathbf{u}| < 5$. This metric measures the time until the solution blows up, not whether there is a particular similarity to the ground truth. As such, a large number of timesteps can be seen as a metric for the autoregressive stability of the network. Figure 20 shows this evaluation for the KS system. The unrolled setups excel at this metric. Unsurprisingly, mitigating the data shift by unrolling the training trajectory stabilizes the inference runs. Through unrolling, the networks were trained on inference-like states. The most stable models were trained with the WIG setup, whose long-term gradients further discourage unphysical outputs that could lead to instabilities in long inference runs. For large networks, their divergence time comes close to the upper threshold we evaluated, i.e. 1000 steps.

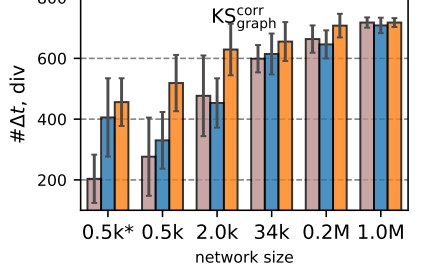 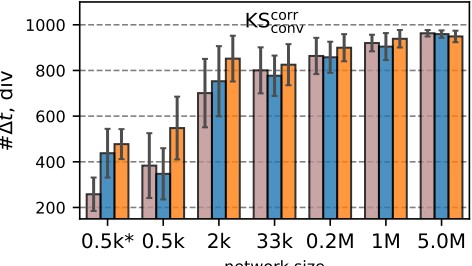

Figure 20: Number of timesteps until the threshold is reached for divergence

## D.5 ADDITIONAL ARCHITECTURE

We tested the generalization of our findings toward attention-gated U-nets (Oktay et al., 2018). The implementation is a 1D version of the network used in the AERO cases introduced in Appendix C.4. Crucially, it features attention gates in its skip connections allowing the network to efficiently mix

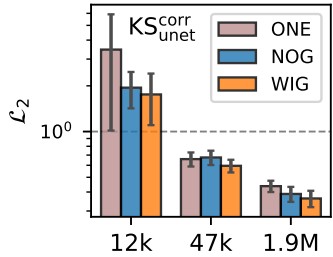

Figure 21: Comparison of $\mathcal{L}_2$ errors for the U-Net architecture on the KS system

global and local structures in the output. The $\mathcal{L}_2$ errors of this architecture on the KS system are visualized in Figure 21. The behavior of ONE, NOG, and WIG matches well with other architectures (i.e. conv and graph nets) studied in the main section. We still observe the best performance with fully differentiable WIG unrolling, while NOG offers slimmer benefits over ONE. We can conclude that our observations regarding the positive effects of unrolling with and without gradients transfer to attention-based networks.

## E    INFERENCE VISUALIZATIONS

This section visualizes typical inference trajectories on which our evaluations were based. Due to the vast amount of trained models, only a very small subset of all models are visualized. The goal is to contextualize the results from the main paper and previous appendix evaluations. The following figures show various training modalities for a fixed initialization, which was randomly selected from our trained models.

Figures 22 and 23 show the trajectory for a graph network in a correction task on the KS system. Both visualizations are from the extrapolative test case with $\mathcal{X} = 4.8$ and $\mathcal{X} = 11.2$ respectively. These characteristic numbers are outside of the lower and upper bounds of the training regime. The figures show the difference from the target state as $u - \tilde{u}$. This error was evaluated for the full range of model sizes as introduced in appendix C and listed in table 2. Figures 24 and 25 similarly display the inference performance of a convolutional model. For both architectures, a clear trend of error reduction is visible for increasing model sizes. At the same time, unrolling reduces errors for a fixed size since NOG and WIG have smaller amplitudes.

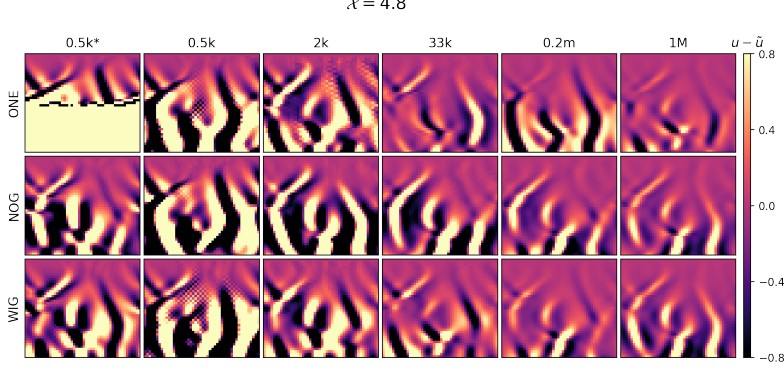

Figure 22:  Inference error trajectory visualization of the GCN correction on the KS system for extrapolation on low $\mathcal{X}$

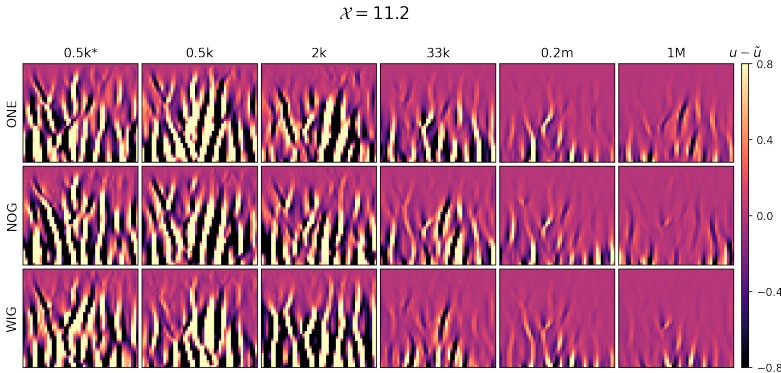

Figure 23: Inference error trajectory visualization of the GCN correction on the KS system for extrapolation on high $\mathcal{X}$

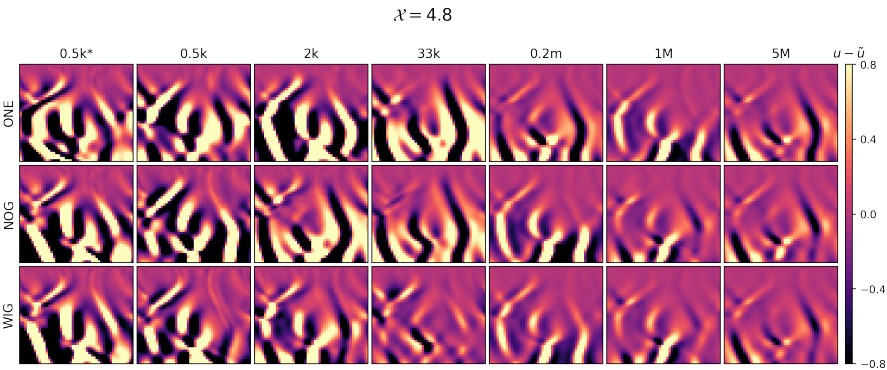

Figure 24: Inference error trajectory visualization of the CNN correction on the KS system for extrapolation on low $\mathcal{X}$

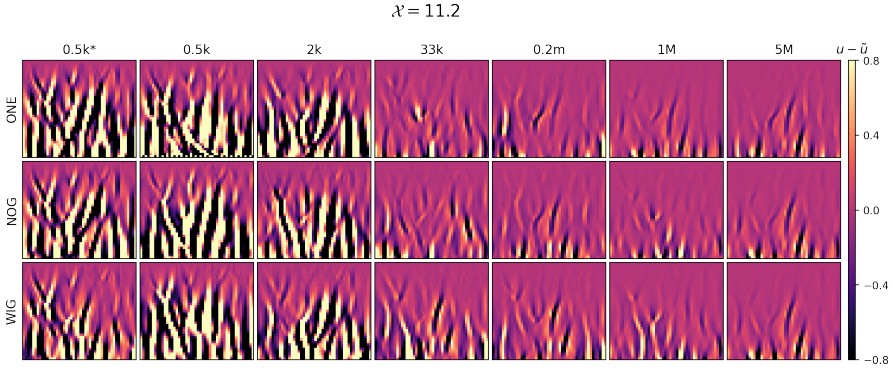

Figure 25: Inference error trajectory visualization of the CNN correction on the KS system for extrapolation on high $\mathcal{X}$

For the KOLM system, we visualize the differences in the vorticity of the predicted state and the ground truth vorticity as $\omega - \tilde{\omega}$ with $\omega = \nabla \times \mathbf{u}$. Again, a fixed initialization was randomly selected from our trained models. The full range of network sizes, as listed in table 3, was visualized. The Vorticity data of the inference frames after 250 steps are plotted next to the reference data in figures 26 and 27 for intermediate and high Reynolds numbers respectively. Similarly, we show interpolative test errors in figure 28 for Re=600 and extrapolative tests in figure 29 on Re=1000.

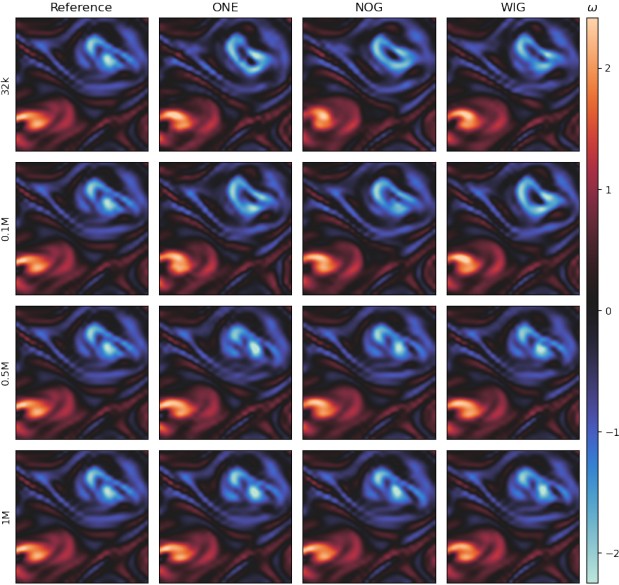

Figure 26: Inference vorticity after 250 steps of the CNN correction on the KOLM system for interpolation on intermediate Re=600, reference data is shown in the left column

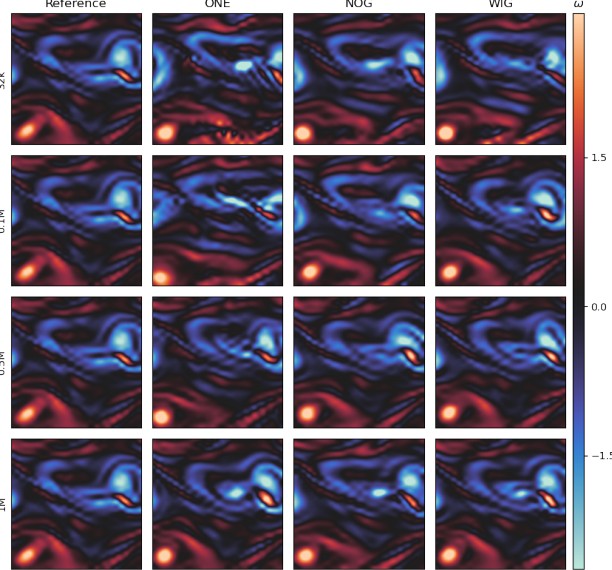

Figure 27: Inference vorticity after 250 steps of the CNN correction on the KOLM system for extrapolation on high Re=1000, reference data is shown in the left column

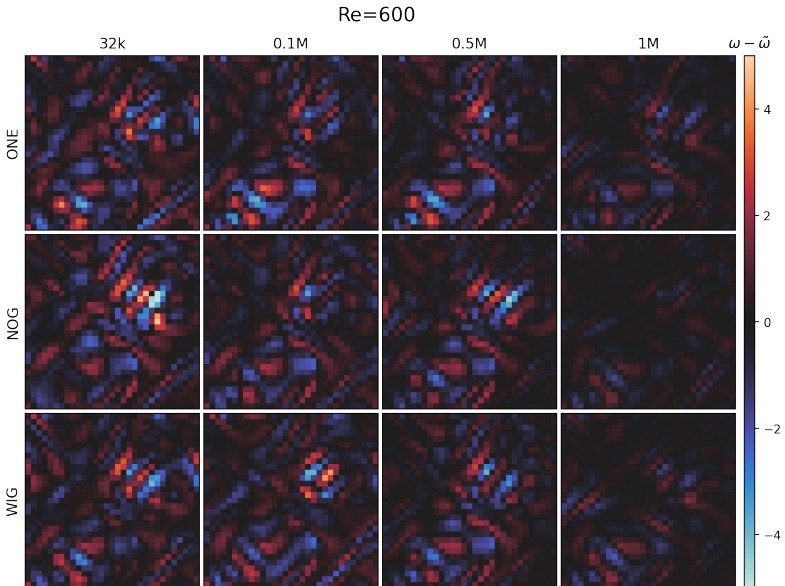

Figure 28: Inference error after 250 steps of the CNN correction on the KOLM system for interpolation on intermediate Re=600, visualized as vorticity

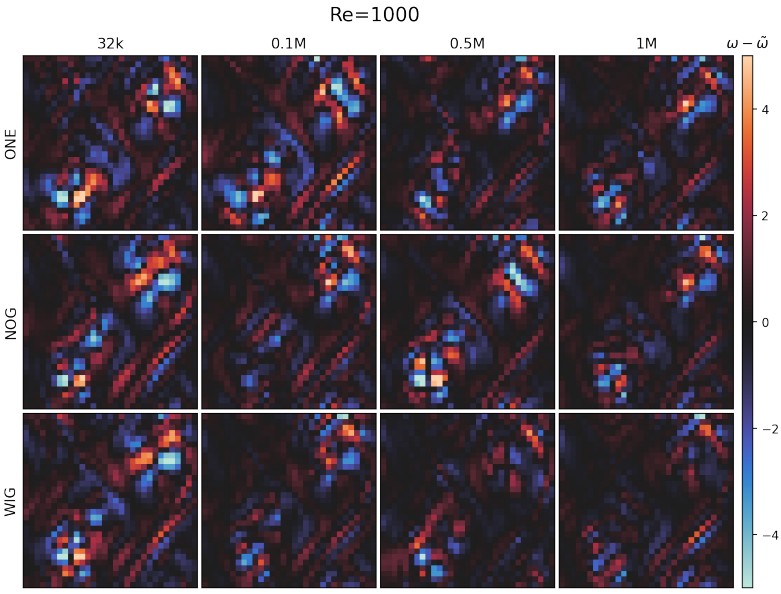

Figure 29: Inference error after 250 steps of the CNN correction on the KOLM system for extrapolation on high Re=1000, visualized as vorticity

## F  EVALUATION DATA

Herein, we provide the statistical data used to generate the figures in the main paper. The error data shown is always gathered from evaluations on a combined test set from interpolative and extrapolative cases, as described in appendix C. For each initial frame in the test set, a trajectory is computed and errors with respect to the ground truth are accumulated, yielding one scalar error value per trajectory. Means and standard deviations are now calculated across these trajectories on a per-model-size and per-training-modality basis. Thus, the tables show these statistics for various model sizes, which are listed for the KS system in table 2, for the wake system in section C.2, for

the KOLM case in table 3, and for the AERO case in table 4. We denote the mean with an overline, e.g. as $\overline{\text{ONE}}$, and the standard deviation as $\sigma$. Additionally, we perform a one-sided Welch's t-test to the statistical significance of the empirical tests and their resulting distributions. For distributions with different means and standard deviations, Welch (1947) defines a t-test as

$$t = \frac{|\overline{X}_1 - \overline{X}_2|}{\sqrt{\dfrac{\sigma_1^2}{M_1} + \dfrac{\sigma_2^2}{M_2}}}. \tag{19}$$

This holds for two distributions with mean $\overline{X}_i$, standard deviation $\sigma_i$, and sample sizes $M_i$. The p-value for a one-sided significance test is then computed as

$$p = Pr(T \leq t|H_0), \tag{20}$$

giving us the probability that the null-hypotheses (i.e. both distributions are identical) is true.

The differences in the studied training modalities are statistically significant for smaller network setups. As model sizes increase, the distribution of trained models becomes more similar. Our heavily overparameterized networks (e.g. 1.0M parameters for 48 degree of freedom KS system) all show highly accurate and stable predictions (see divergence time metric, Figure 20 Appendix C). Based on the theoretical considerations above, this means that the attractor of the learned dynamics is closely aligned with the ground truth system. These setups thus display less data shift and the benefits of unrolling are reduced. When these conditions in the overparameterized regime are met, NOG models are at a disadvantage due to their mismatch between gradients and loss landscape. However, these heavily oversized architectures are not practically relevant for scientific computing due to their weak scaling compared to numerical approaches. This evaluation confirms that for relevant, small to medium sized networks our results are statistically significant and hence that conclusions can be drawn.

The tables relate to the figures as follows:

- Figure 4 visualizes tables 5, 7, 8
- Figure 6 visualizes tables 13, 14, 15
- Figure 7 visualizes table 16
- Figure 9 visualizes tables 10, 11, 9
- Figure 11 visualizes tables 17, 18
- Figure 12 visualizes table 12

Table 5: Correction GCN $\mathcal{L}_2$ errors on the KS system

|  | $\overline{\text{ONE}}$ | $\overline{\text{NOG}}$ | $\overline{\text{WIG}}$ | $\sigma(\text{ONE})$ | $\sigma(\text{NOG})$ | $\sigma(\text{WIG})$ | $p_{\text{ONE}}^{\text{NOG}}$ | $p_{\text{ONE}}^{\text{WIG}}$ |
|---|---|---|---|---|---|---|---|---|
| 0.5k* | 2.67063 | 2.07773 | 1.32860 | 0.61222 | 0.62611 | 0.09505 | 0.00220 | 0.00000 |
| 0.5k | 3.42429 | 2.01830 | 1.15905 | 2.00918 | 0.81870 | 0.24652 | 0.00310 | 0.00001 |
| 2k | 2.05080 | 1.53242 | 0.77671 | 1.58849 | 1.07397 | 0.20318 | 0.11706 | 0.00051 |
| 33k | 0.42879 | 0.39406 | 0.33274 | 0.10139 | 0.11152 | 0.10516 | 0.15465 | 0.00278 |
| 0.2M | 0.18810 | 0.20359 | 0.16093 | 0.04164 | 0.04815 | 0.05224 | 0.14164 | 0.03843 |
| 1M | 0.12671 | 0.16176 | 0.12054 | 0.01942 | 0.03265 | 0.02622 | 0.00010 | 0.20174 |

Table 6: Correction CNN $\mathcal{L}_2$ errors on the KS system

|       | $\overline{\text{ONE}}$ | $\overline{\text{NOG}}$ | $\overline{\text{WIG}}$ | $\sigma(\text{ONE})$ | $\sigma(\text{NOG})$ | $\sigma(\text{WIG})$ | $p_{\text{ONE}}^{\text{NOG}}$ | $p_{\text{ONE}}^{\text{WIG}}$ |
|-------|---------|---------|---------|---------|---------|---------|---------|---------|
| 0.5k* | 2.79276 | 1.82674 | 1.36738 | 0.53627 | 0.33042 | 0.18318 | 0.00000 | 0.00000 |
| 0.5k  | 2.04899 | 1.77093 | 1.32922 | 0.76169 | 0.47778 | 0.46564 | 0.08736 | 0.00045 |
| 2k    | 1.04701 | 0.97612 | 0.80260 | 0.16266 | 0.10389 | 0.08582 | 0.05436 | 0.00000 |
| 33k   | 0.39871 | 0.42818 | 0.34545 | 0.08113 | 0.08503 | 0.05931 | 0.13464 | 0.01148 |
| 0.2M  | 0.23991 | 0.22823 | 0.19139 | 0.06584 | 0.05224 | 0.03468 | 0.26910 | 0.00296 |
| 1M    | 0.12554 | 0.16273 | 0.09885 | 0.01664 | 0.06723 | 0.01865 | 0.01067 | 0.00001 |
| 2M    | 0.09592 | 0.09519 | 0.07090 | 0.01241 | 0.01160 | 0.00889 | 0.42393 | 0.00000 |

Table 7: Correction CNN $\mathcal{L}_2$ errors on the WAKE system

|      | $\overline{\text{ONE}}$ | $\overline{\text{NOG}}$ | $\overline{\text{WIG}}$ | $\sigma(\text{ONE})$ | $\sigma(\text{NOG})$ | $\sigma(\text{WIG})$ | $p_{\text{ONE}}^{\text{NOG}}$ | $p_{\text{ONE}}^{\text{WIG}}$ |
|------|---------|---------|---------|---------|---------|---------|---------|---------|
| 6.3k | 0.12045 | 0.07038 | 0.05515 | 0.02353 | 0.01374 | 0.00990 | 0.00001 | 0.00000 |
| 67k  | 0.07641 | 0.04430 | 0.03663 | 0.03321 | 0.01106 | 0.00568 | 0.00476 | 0.00076 |
| 1M   | 0.03389 | 0.02575 | 0.02496 | 0.00724 | 0.00733 | 0.00717 | 0.01120 | 0.00632 |

Table 8: Correction CNN $\mathcal{L}_2$ errors on the KOLM system

|       | $\overline{\text{ONE}}$ | $\overline{\text{NOG}}$ | $\overline{\text{WIG}}$ | $\sigma(\text{ONE})$ | $\sigma(\text{NOG})$ | $\sigma(\text{WIG})$ | $p_{\text{ONE}}^{\text{NOG}}$ | $p_{\text{ONE}}^{\text{WIG}}$ |
|-------|---------|---------|---------|---------|---------|---------|---------|---------|
| 32k   | 0.04097 | 0.04140 | 0.03645 | 0.00833 | 0.00506 | 0.00398 | 0.45147 | 0.09404 |
| 0.1M  | 0.03064 | 0.02576 | 0.02504 | 0.00344 | 0.00511 | 0.00242 | 0.02090 | 0.00104 |
| 0.5Mk | 0.01635 | 0.01394 | 0.01420 | 0.00172 | 0.00144 | 0.00201 | 0.00444 | 0.01878 |
| 1M    | 0.01046 | 0.00859 | 0.00820 | 0.00084 | 0.00110 | 0.00122 | 0.00093 | 0.00035 |

Table 9: Prediction CNN $\mathcal{L}_2$ errors on the WAKE system

|      | $\overline{\text{ONE}}$ | $\overline{\text{NOG}}$ | $\overline{\text{WIG}}$ | $\sigma(\text{ONE})$ | $\sigma(\text{NOG})$ | $\sigma(\text{WIG})$ | $p_{\text{ONE}}^{\text{NOG}}$ | $p_{\text{ONE}}^{\text{WIG}}$ |
|------|---------|---------|---------|---------|---------|---------|---------|---------|
| 6.3k | 2307.82003 | 0.12045 | 0.11758 | 4614.57276 | 0.01364 | 0.01212 | 0.06560 | 0.06560 |
| 67k  | 0.15232 | 0.06248 | 0.06518 | 0.06279 | 0.00593 | 0.00545 | 0.00014 | 0.00018 |
| 1M   | 0.03336 | 0.03540 | 0.02791 | 0.00261 | 0.01500 | 0.00084 | 0.33794 | 0.00000 |

Table 10: Prediction GCN $\mathcal{L}_2$ errors on the KS system

|       | $\overline{\text{ONE}}$ | $\overline{\text{NOG}}$ | $\overline{\text{WIG}}$ | $\sigma(\text{ONE})$ | $\sigma(\text{NOG})$ | $\sigma(\text{WIG})$ | $p_{\text{ONE}}^{\text{NOG}}$ | $p_{\text{ONE}}^{\text{WIG}}$ |
|-------|---------|---------|---------|---------|---------|---------|---------|---------|
| 0.5k* | 7.98357 | 11.63567 | 10.23652 | 2.75299 | 1.57462 | 2.53653 | 0.00000 | 0.00526 |
| 0.5k  | 18.06462 | 5.10212 | 9.51381 | 5.02841 | 3.35803 | 4.80894 | 0.00000 | 0.00000 |
| 2k    | 10.41816 | 2.89637 | 5.62151 | 5.18653 | 0.92519 | 2.53810 | 0.00000 | 0.00033 |
| 33k   | 4.16912 | 1.44542 | 1.77320 | 2.10251 | 0.10168 | 0.68453 | 0.00000 | 0.00001 |
| 0.2M  | 1.13015 | 0.91450 | 0.79421 | 0.05839 | 0.08480 | 0.07173 | 0.00000 | 0.00000 |
| 1M    | 0.88423 | 0.81924 | 0.71090 | 0.05668 | 0.06682 | 0.07325 | 0.00101 | 0.00000 |

Table 11: Prediction CNN $\mathcal{L}_2$ errors on the KS system

|       | $\overline{\text{ONE}}$ | $\overline{\text{NOG}}$ | $\overline{\text{WIG}}$ | $\sigma(\text{ONE})$ | $\sigma(\text{NOG})$ | $\sigma(\text{WIG})$ | $p_{\text{ONE}}^{\text{NOG}}$ | $p_{\text{ONE}}^{\text{WIG}}$ |
|-------|---------|---------|---------|---------|---------|---------|---------|---------|
| 0.5k* | 19.87167 | 13.16589 | 17.09354 | 0.60110 | 2.18501 | 1.16960 | 0.00000 | 0.00000 |
| 0.5k  | 6.63006 | 4.37441 | 6.78108 | 3.99266 | 2.30527 | 2.03592 | 0.01744 | 0.44051 |
| 2k    | 2.51767 | 2.34324 | 2.73029 | 0.47555 | 0.34352 | 0.97282 | 0.09577 | 0.19269 |
| 33k   | 1.38413 | 1.49134 | 1.38093 | 0.18477 | 0.20430 | 0.15411 | 0.04493 | 0.47643 |
| 0.2M  | 1.02191 | 0.97444 | 0.87086 | 0.18962 | 0.12227 | 0.05520 | 0.17634 | 0.00075 |
| 1M    | 0.76798 | 0.85323 | 0.64513 | 0.06914 | 0.12842 | 0.04222 | 0.00638 | 0.00000 |

Table 12: Prediction Unet $\mathcal{L}_2$ errors on the AERO system

|  | $\overline{\text{ONE}}$ | $\overline{\text{NOG}}$ | $\overline{\text{WIG}}$ | $\sigma(\text{ONE})$ | $\sigma(\text{NOG})$ | $\sigma(\text{WIG})$ |
|---|---|---|---|---|---|---|
| 0.5M | 0.05046 | 0.03143 | 0.02975 | 0.01302 | 0.00420 | 0.00221 |
| 2M | 0.03302 | 0.02815 | 0.02829 | 0.00251 | 0.00090 | 0.00106 |
| 8M | 0.02892 | 0.02392 | 0.02388 | 0.00505 | 0.00153 | 0.00204 |

Table 13: Correction CNN $\mathcal{L}_2$ errors on KS system for multiple unrollings

|  | $\overline{\text{NOG}}$ | $\overline{\text{WIG}}$ | $\sigma(\text{NOG})$ | $\sigma(\text{WIG})$ |
|---|---|---|---|---|
| m=2 | 0.34026 | 0.33344 | 0.06799 | 0.07425 |
| m=3 | 0.33358 | 0.32817 | 0.07953 | 0.05533 |
| m=4 | 0.32845 | 0.31876 | 0.06794 | 0.05780 |
| m=5 | 0.36300 | 0.29802 | 0.06534 | 0.06446 |
| m=6 | 0.34506 | 0.29201 | 0.05682 | 0.06101 |
| m=8 | 0.36041 | 0.28614 | 0.04762 | 0.04553 |
| m=10 | 0.39730 | 0.28358 | 0.07861 | 0.06345 |
| m=12 | 3.30768 | 0.26729 | 8.65391 | 0.07201 |
| m=14 | inf | 0.25408 | 0.00000 | 0.06418 |
| m=16 | inf | 0.24387 | 0.00000 | 0.05274 |
| m=18 | inf | 0.25749 | 0.00000 | 0.05718 |
| m=20 | inf | 3.46048 | 0.00000 | 9.12454 |

Table 14: Correction CNN $\mathcal{L}_2$ errors on KOLM system for multiple curriculums

|  | $\overline{\text{NOG}}$ | $\overline{\text{WIG}}$ | $\sigma(\text{NOG})$ | $\sigma(\text{WIG})$ |
|---|---|---|---|---|
| 1-2-4 | 0.03064 | 0.01635 | 0.00344 | 0.00172 |
| 2-2-4 | 0.01046 | 0.04140 | 0.00084 | 0.00506 |
| 4-4-4 | 0.02576 | 0.01394 | 0.00511 | 0.00144 |

Table 15: Correction CNN $\mathcal{L}_2$ errors on KOLM system when trained without learning rate schedules

|  | $\overline{\text{ONE}}$ | $\overline{\text{NOG}}$ | $\overline{\text{WIG}}$ | $\sigma(\text{ONE})$ | $\sigma(\text{NOG})$ | $\sigma(\text{WIG})$ |
|---|---|---|---|---|---|---|
| 32k | 0.04860 | 0.04563 | 0.05165 | 0.00651 | 0.00528 | 0.00626 |
| 0.1M | 0.03478 | 0.03701 | 0.03708 | 0.00271 | 0.00689 | 0.00433 |
| 0.5M | 0.03090 | 0.02827 | 0.03057 | 0.00267 | 0.00320 | 0.00332 |
| 1M | 0.02935 | 0.02244 | 0.02635 | 0.00965 | 0.00398 | 0.00394 |

Table 16: Correction CNN $\mathcal{L}_2$ errors on KS system for gradient stopping variants

|  | m=6 | m=6$_{\text{init}}$ | w=1 | w=2 | v=1 | v=2 |
|---|---|---|---|---|---|---|
| $\overline{\text{WIG}}$ | 0.388526 | 0.281911 | 0.374733 | 0.396608 | 0.441374 | 0.407062 |
| $\sigma(\text{WIG})$ | 0.092057 | 0.050417 | 0.097563 | 0.102014 | 0.156538 | 0.101093 |

Table 17: GCN $\mathcal{L}_2$ errors on KS system when transitioning from prediction to correction

|     | $\overline{\text{ONE}}$ | $\overline{\text{NOG}}$ | $\overline{\text{WIG}}$ | $\sigma(\text{ONE})$ | $\sigma(\text{NOG})$ | $\sigma(\text{WIG})$ |
| --- | --- | --- | --- | --- | --- | --- |
| 10% | 1.46963 | 1.15836 | 1.04210 | 0.27693 | 0.09666 | 0.15044 |
| 20% | 1.22752 | 1.06870 | 0.92173 | 0.16561 | 0.11952 | 0.05283 |
| 30% | 1.05448 | 1.13224 | 0.93338 | 0.17234 | 0.18949 | 0.10398 |
| 40% | 0.98373 | 0.92035 | 0.77333 | 0.16078 | 0.12198 | 0.09403 |
| 50% | 0.83278 | 0.78407 | 0.75527 | 0.11489 | 0.12352 | 0.11875 |
| 60% | 0.73871 | 0.70152 | 0.61906 | 0.14414 | 0.09365 | 0.06889 |
| 70% | 0.65280 | 0.61808 | 0.46474 | 0.10726 | 0.09699 | 0.07094 |
| 80% | 0.46519 | 0.42852 | 0.39318 | 0.10353 | 0.10320 | 0.13866 |
| 90% | 0.26651 | 0.15659 | 0.18686 | 0.09774 | 0.01506 | 0.06874 |

Table 18: CNN $\mathcal{L}_2$ errors on KS system when transitioning from prediction to correction

|     | $\overline{\text{ONE}}$ | $\overline{\text{NOG}}$ | $\overline{\text{WIG}}$ | $\sigma(\text{ONE})$ | $\sigma(\text{NOG})$ | $\sigma(\text{WIG})$ |
| --- | --- | --- | --- | --- | --- | --- |
| 10% | 1.08743 | 1.07423 | 1.00705 | 0.10005 | 0.08438 | 0.09433 |
| 20% | 0.98619 | 0.97025 | 0.85021 | 0.12132 | 0.10936 | 0.07485 |
| 30% | 0.89545 | 0.87275 | 0.82945 | 0.07255 | 0.10187 | 0.07454 |
| 40% | 0.81295 | 0.81708 | 0.72305 | 0.13808 | 0.10260 | 0.06438 |
| 50% | 0.70735 | 0.71738 | 0.66283 | 0.09532 | 0.08907 | 0.07525 |
| 60% | 0.55892 | 0.60302 | 0.55926 | 0.12181 | 0.10013 | 0.05164 |
| 70% | 0.45414 | 0.45675 | 0.48273 | 0.12301 | 0.11392 | 0.12604 |
| 80% | 0.33173 | 0.28746 | 0.31830 | 0.08661 | 0.06363 | 0.07834 |
| 90% | 0.14203 | 0.11449 | 0.12929 | 0.02205 | 0.01993 | 0.04566 |

