# OpenReview forum: "How Temporal Unrolling Supports Neural Physics Simulators"
_ICLR.cc/2024/Conference — Submitted to ICLR 2024_

### Official Review · Reviewer_BVZT · 2023-10-30

**Soundness:** 4 excellent
**Presentation:** 3 good
**Contribution:** 3 good
**Rating:** 6
**Confidence:** 2

**Summary:**

This paper studies the sources of error in training neural networks for prediction or correction tasks in PDE simulations.  The authors compare three different training paradigms which are ONE (training only by simulating one step forward at a time and evaluating the error after this step), NOG (to train with sequences of length m but to have gradients only flow backward a single step while training) and WIG (use full back propagation through the m steps of simulation time). Unsurprisingly, the WIG method works the best on a suite of tasks, but in many cases NOG can perform almost as well. While the NOG and MIG methods get slightly better with m > 1 , the NOG and MIG methods can be sensitive for large m due to data shift and exploding gradients respectively. Surprisingly the authors report a much more systematic trend with respect to model size: they report a scaling law for simulation accuracy which falls as a power law with model parameter count.

**Strengths:**

This paper studies an important problem of using deep networks as simulation/prediction engines for PDE simulation. By examining the difference between these three training paradigms, the authors were able to disentangle the effects of data distribution shift and backpropagation errors.  This allows one to evaluate which types of errors are more important in simulation settings of interest including several popular PDEs. The set of experiments is quite extensive with many additional tests in the appendix.

**Weaknesses:**

Some concepts are introduced without explicit definition. I think that the NOG + MIG could be explained more clearly, specifically giving a longer explanation of how the gradients are computed and defining what the authors mean by “loss landscape changes” and “data distribution shifts.”  I think I understand what the authors mean, but it could be helpful to have this more explicitly explained somewhere.

**Questions:**

1. Based on the provided experimental results would the authors conclude that the scale of the model is more important in determining good performance than the specific error scheme?
2. Why do the authors think that learning rate schedules or unrolling curricula are so important for stability in these settings?
3. How does the compute of the best performing deep learning models compare to direct solvers?

---

> ### Author Response · Authors · 2023-11-20
> **Response to Reviewer BVZT - Part 1/2**
>
> Thank you for the review. Please find individual responses to all of your concerns and questions below.
>
> ### Weaknesses
>
> *Some concepts are introduced without explicit definition. I think that the NOG + MIG could be explained more clearly, specifically giving a longer explanation of how the gradients are computed and defining what the authors mean by “loss landscape changes” and “data distribution shifts.”*
>
> Thank you for the comment. We have now added a theoretical section in the general response that formally introduces the data shift concept, and should help clarify your concerns. Furthermore, appendix A has a detailed explanation of gradient calculation for NOG and WIG setups. The theoretical analysis then expands on these and studies the loss-gradient mismatch in NOG unrolling.
>
> The revised paper will also motivate the NOG setup more clearly. Crucially, no-gradient unrolling could be achieved by interfacing existing numerical code with a machine-learning library. It is a viable option and first step for scientific domains that are dominated by known, complex, and validated code bases, like computational fluid dynamics.
>
> ### Questions
>
> *Q1: Based on the provided experimental results would the authors conclude that the scale of the model is more important in determining good performance than the specific error scheme?*
>
> Network size does indeed significantly influence accuracy. Our results actually provide insights beyond this simple fact: We provide convergence rates of the inference error with respect to the parameter count for machine-learned correction and prediction methods. For the physical systems, numerical solvers, and network architectures studied, these convergence rates are small in comparison to purely numerical approaches. A grid refinement of the underlying solver might thus yield better results than an extension of the neural network parameter space for the same additional computational burden. Based on this observation, we conclude that slimmer architectures are of greater interest to the scientific computing community.
> We believe that this is an important and novel consideration that goes beyond previous studies.

---

> ### Author Response · Authors · 2023-11-20
> **Response to Reviewer BVZT - Part 2/2**
>
> ### Questions
>
> *Q2: Why do the authors think that learning rate schedules or unrolling curricula are so important for stability in these settings.*
>
> Training with a curriculum avoids the stiffness of long unrolling horizons in the early stages of the network optimization. Starting the training with a low unrolling number (i.e. m=1) and progressively increasing this horizon ensures that the network produces physical outputs once long horizons are reached. Once the network produces outputs on long horizons that are closer to the target data, these long horizons can produce meaningful learning signals. If we trained without a curriculum, the gradients by later steps in the unrolling would produce very noisy gradients for networks without pretraining. This effect exists for NOG and WIG training. It is further amplified by the WIG backpropagation due to the chaotic dynamics of the underlying system. This can be taken from our added theoretical analysis where we find that the long-term part of the gradients, which differentiates NOG from WIG, grows over long horizons in relation to the NOG gradient. Our curriculum studies in Figure 6 support this observation, where we can see that NOG suffers less from training without a curriculum.
>
> Learning rate schedules have a regularizing effect on the network updates computed during optimization. When the unrolling horizon is increased due to curriculum learning, the output states deviate stronger from the ground truth data for later steps. This can make the learning signal unstable. The learning rate schedule is introduced to counteract this effect. It is thus a direct consequence of training with a curriculum.
>
> ---
>
> *Q3: How does the compute of the best performing deep learning models compare to direct solvers?*
>
> Thank you for this comment, it touches upon one central insight of our work. We have established convergence rates for network architectures and seen how their scaling motivates slimmer sizes over overparametrized variants. This is also true in our tests, where large models require substantially more operations than numerical solvers, which translates to the computing time. We also include convergence studies for our datasets and network architectures. A rate of $n^{⅓}$ can be fit to the network's convergence rate, where n represents the number of network parameters. In practice, this means that a doubling in the computational complexity only yields an accuracy improvement factor of 1.26. This convergence rate is poor in comparison to numerical approaches, where rates larger than $n^2$ are the standard (i.e. improvement of 4x for a doubling in numerical gridpoints). This observation is crucial to the scientific machine-learning community and has often been overlooked in previous studies.
>
> At the same time, the goal of our work is not to propose a specific architecture that outperforms classical approaches. Rather, we compare strategies that ultimately help train such models as well as extract and study the relevant quantities in these training strategies.

---

### Official Review · Reviewer_CxuR · 2023-10-30

**Soundness:** 3 good
**Presentation:** 3 good
**Contribution:** 2 fair
**Rating:** 5
**Confidence:** 4

**Summary:**

The paper attempts to disentangle the possible benefits of temporal unrolling for training neural PDE solvers. The authors focus their investigations on two possibilities: (1) unrolling improves simulators by increasing their resilience to distribution shift when applied over many time steps (2) propagating gradient information through the many steps of the simulation leads to better modeling of long-term interactions. With experiments on several non-linear chaotic systems the authors show that unrolling without gradients can lead to significant improvements over one-step evolution but that unrolling with gradients leads to even greater improvements, especially over longer horizons (e.g. 8 or greater). However, despite notable difference between these two approaches to unrolling, simply increasing the parameter count of the model often has an equivalent or greater impact on the ultimate test loss.

**Strengths:**

I really admire the goals of this paper. Using physics-inspired neural networks often requires many subtle but important design decisions, and the consequences of the decisions aren't always clear. The authors do a good job of isolating an important and interesting research question and present results for several key ablations on a few popular datasets. The results themselves are fairly interesting, as they suggest, somewhat counterintuitively, that back-propagating through the unrolling process if sometimes unnecessary or at least of minimal marginal benefit. Likewise, in other cases, the results suggest that gradients have an extremely significant effect on the ability of the training to converge when simulating over long horizons.

**Weaknesses:**

While I think the goals of the paper are good, when I finished reading it, I wasn't sure exactly what to take away from the experience. Is the primary take-away that gradients are largely unnecessary if training is conducted with an appropriate curriculum? Is this primarily significant because it makes the training process easier for practitioners in some way and thereby facilitates scaling to larger models and systems?  It's not totally clear how much the proposed ablations matter when the effect of model scaling seemed to significantly outweigh the investigated effects of unrolling, so I'm left wondering what the practical impact is.

Novelty and significance: Does this paper have any prescriptions that go beyond prior work, or is it simply a description of ablating several methods from prior work on a collection of datasets and model sizes? Could you turn any observations into an improved method that would outperform an established baseline? There are essentially no baselines in this paper, so I have no sense for how the effect of the ablations compare to the effect of swapping methods.

Depth of insight: There does seem to be some indication that gradients are very helpful in some cases and but not in others, but the authors don't dig deeper into this observation. What is captured my the multi-step gradients that might not be captured by training without gradient on data derived from unrolling (data that captures the distribution shift)? You note that a curriculum is necessary to scale learning to longer rollouts. How would you explain the observation in Figure 6 (left) that gradients aid convergence on longer rollouts whereas training without gradients completely fails? Is one possible explanation that gradients lead to a form of reweighting on the earlier steps in the trajectory, giving them more significance in the learning process, whereas training without gradients on long rollouts inadvertently upweights later parts of the trajectory before the model has learned to fit the beginning of the trajectory well? Could such observations be converted into a method that provably performs better than existing methods. The authors offer a few concrete take-aways, but they don't seem to significantly expand upon insights/methods from prior work.

**Questions:**

I've provided a few questions in my response in the weaknesses section that could start a valuable discussion.

---

> ### Author Response · Authors · 2023-11-20
> **Response to Reviewer CxuR - Part 1/2**
>
> Thank you for the review. Please find individual responses to all of your concerns and questions below.
>
> ### Weaknesses
>
> *W1: Is the primary take-away that gradients are largely unnecessary if training is conducted with an appropriate curriculum? Is this primarily significant because it makes the training process easier for practitioners in some way and thereby facilitates scaling to larger models and systems?*
>
> Thank you for this comment. It goes to the essence of our paper, where we aimed to go beyond the observation that model size dominates the test loss. While this effect is apparent in our data, we also include convergence studies for our datasets and network architectures. A rate of $n^{⅓}$ can be fit to the network's convergence rate, where n represents the number of network parameters. In practice, this means that a doubling in the computational complexity only yields an accuracy improvement factor of 1.26. This convergence rate is poor in comparison to numerical approaches, where rates larger than $n^2$ are the standard (i.e. improvement of 4x for a doubling in numerical gridpoints). This observation is crucial to the scientific machine-learning community and has often been overlooked in previous studies. Applications in computational sciences can thus profit from using hybrids between numerical and neural architecture. This combines the superior scaling of numerical solvers with the intrinsic benefits of neural networks. These benefits include reduced modeling bias, observation-based data fitting, and flexibility of application. Since these can be utilized regardless of size, slimmer architectures will be favorable based on the convergence rates above. With this in mind, we can now consider our evaluations of the small to mid-sized networks. Here, the studied training modalities showed significantly different test performances, where with-gradient unrolling performed best and no-gradient unrolling came second. For practitioners in the scientific computing community, the conclusions are twofold:
>
> (1) Unrolling improves the accuracy of a machine-learning augmented simulator. This is true even for a no-gradient training. Crucially, no-gradient unrolling could be achieved by interfacing existing numerical code with a machine-learning library. It is a viable option and first step for scientific domains that are dominated by known, complex, and validated code bases, like computational fluid dynamics.
>
> (2) For best results, a differentiable setup is still required. Our results allow an estimation of the expected return of implementing either an interface to existing numerical solvers (no-gradient) or a differentiable solver (with-gradient).
>
> Additionally, we conducted studies on curriculum learning and learning rate scheduling. Effective unrolled training relies on both of these strategies for stable optimization. This should stress that the mentioned performance improvements can only be achieved by carefully designing the training setup and should help avoid this pitfall in future studies.
> We have revised the paper in section 1 (parameter count) and section 5.1 (size) and believe that the viewpoint provided above is now clearer.
>
> ---
>
> *W2.1: Novelty and significance: Does this paper have any prescriptions that go beyond prior work, or is it simply a description of ablating several methods from prior work on a collection of datasets and model sizes? Could you turn any observations into an improved method that would outperform an established baseline?*
>
> To our knowledge, the no-gradient (NOG) approach has not been studied in previous work. While with-gradient unrolling was shown to be beneficial in previous papers, our results show how these benefits are exhibited in no-gradient unrolling. As pointed out above, the NOG training modality constitutes an attractive option for merging machine learning with scientific computing. Our experiments thus form a basis for motivating this process and help estimate its expected gains. For many applications, differentiable simulators are not available (e.g. boundary-fitted Navier Stokes, two-fluid flows). When one considered machine-learning these problems, only pure predictions were possible. Our results indicate that a 5x accuracy boost could be achieved by integrating low-fidelity physics priors in a simple NOG setup. This estimate can be taken by comparing KS predictions (WIG) to corrections (NOG) in Tables 5&7 for GCNs and 6&8 for CNNs. Such accuracy boosts are achievable in many settings by interfacing existing code bases with machine learning.

---

> ### Author Response · Authors · 2023-11-20
> **Response to Reviewer CxuR - Part 2/2**
>
> ### Weaknesses
>
> *W2.2: There are essentially no baselines in this paper, so I have no sense for how the effect of the ablations compare to the effect of swapping methods.*
>
> We trained a broad range of architectures that cover more than 3 orders of magnitude in network size, resulting in over 2000 trained models for this study. This capacity dimension of neural networks is often overlooked, but of critical importance to scientific computing. The range of networks covers common choices in prominent previous work. For instance, the 1.0M parameter graph-network aligns with Brandstetter et al. (2022), while the convolutional network with 250k parameters from Kochkov et al. (2021) is also contained in our capacity range. Similarly, the convolutional network sizes (60k and 250k) studied by Um et al. (2020) fall within the same range. This is also true for the convolutional ResNet architectures in Stachenfeld et al. (2022) with 200k parameters for 1D problems and 580k parameters in 2D.
>
> ---
>
> *W3: What is captured my the multi-step gradients that might not be captured by training without gradient on data derived from unrolling (data that captures the distribution shift)? You note that a curriculum is necessary to scale learning to longer rollouts. How would you explain the observation in Figure 6 (left) that gradients aid convergence on longer rollouts whereas training without gradients completely fails? Could such observations be converted into a method that provably performs better than existing methods?*
>
> Thanks for the detailed questions. A better intuition of the various training modalities can be gained from the theoretical considerations we have added in this revision. They formalize the thoughts that lead to the schematic plot in Figure 1. One take away is that unrolling offers (theoretical) full exploration of the learned dynamics, which reduces the data shift for long unrollings. However, for chaotic dynamics, there are conflicting evolutions for gradient accuracy no matter whether no-gradient or with-gradient calculations are chosen. A mismatch between gradients and loss landscape grows quickly for no-gradient unrolling, leading to reduced performance when longer horizons are chosen. With-gradient unrolling similarly suffers at longer horizons as the chaotic dynamics lead to gradient blow-up for large m. In practice, this means that no-gradient unrolling is only effective on shorter horizons than with-gradient unrolling. Figure 6 confirms this observation. For a variation of unrolling lengths, with-gradient unrolling remains stable for longer while no-gradient unrolling blows up faster. The explanation you have provided sounds intuitive, it is true that no-gradient unrolling discards all long-term feedback and gradients on later steps could thus be meaningless. In our added theoretical analysis, we show how the error term in the no-gradient calculation grows a lot faster than the actual computed contribution when m grows. As such, poor performance of no-gradient training is expected for long unrollings.
>
> You also mention how the benefits of no-gradient training vary over some of our tests. We believe that the explanation goes along another dimension exposed in empirical tests, which is the network size. As the capacity of the network grows, its’ modeled dynamics move closer to the ground truth system. The attractors of the learned system and the ground truth system become more alike the better the learned mapping becomes. Thus, for highly accurate networks, unrolling becomes less crucial to explore the learned attractor. In such cases, the ground truth data is a good enough representation of this state space. The unrolling setups reduce the data shift less effectively. However, no-gradient unrolling maintains the disadvantage of only approximating the true gradient and therefore falls behind in these settings. As elaborated upon in our previous responses, this only is the case for overparameterized networks, which due to their sheer size can hardly be justified over numerical approaches.
>
> One result of our study is that unrolling, its variants, and its effect on training are more complex than initially anticipated in previous work. A one-size-fits-all recommendation is hard to make in light of all the dimensions studied in this paper. Our results do allow for an estimate of potential benefits in situations that are common in scientific computing with machine learning. As outlined before, the inclusion of physical priors via numerical solvers (prediction vs correction), as well as the differences in training with physical priors (ONE-NOG-WIG) and their effect on model performance can be motivated and compared based on our results.

---

### Official Review · Reviewer_9oEF · 2023-11-08

**Soundness:** 2 fair
**Presentation:** 3 good
**Contribution:** 2 fair
**Rating:** 5
**Confidence:** 4

**Summary:**

This paper analyzes the methods of using neural networks to unroll PDEs. Specifically, based on the unrolling steps and differentiability, three variations of methods are tested under different setups. The conclusion is drawn empirically that multi-step differentiable unrolling increases the PDE solution accuracy.

**Strengths:**

The paper presentation is high quality.

The experiment setup is comprehensive, providing varying scenarios to compare different training methods for unrolling PDE.

**Weaknesses:**

The motivation for training with a non-differentiable unrolling method is not very clear.

Some test details are unclear, e.g., the experiment setup behind Figure 1 and the quantification of data shift in different cases. As the paper focuses on benchmarking the training methods, the details are essential for empirical analysis.

The empirical results lack theoretical support. The reviewer doubt if the results are convictive over different physical systems and NN models.

**Questions:**

Figure 1 is intuitive, but how do you get the curves? If it shows the result of a toy example, what is the specific setup, e.g., physical system, NN models, available data?

For correction setup, physics priors are available. The later experiments include changing the prior’s accuracy to show the performance variation. With priors, can physics-informed NN be used to solve the problem? Then, how about comparing the three variants of unrolling training with SOTA physics-informed NN for solving PDE?

The average improvement on different testing scenarios using non-differential and differential unrolling training methods looks promising, but is there any theoretical analysis behind it to support the performance so that it can be generally applicable?

The training of multi-step unrolling seems to be more effective than one-step unrolling. How should the step m be chosen? Figure 6 shows the results with different m values in the Kuramoto-Sivashinsky equation. How about the other systems? Will the step size be decided from validation?

---

> ### Author Response · Authors · 2023-11-20
> **Response to Reviewer 9oEF**
>
> Thank you for the review. Please find individual responses to all of your concerns and questions below.
>
> ### Weaknesses
>
> *W1: The motivation for training with a non-differentiable unrolling method is not very clear.*
>
> We thank the reviewer for the comment. Differentiability is generally not given in codebases used for scientific computing. These codebases are the product of long development and validation processes, where establishing differentiability would be a tremendous effort. The no-gradient setup provides a very attractive strategy that would only require an interface between a numerical simulator and the machine learning augmentation. Reliable statistical evaluations are necessary to estimate the efficacy of these training modalities and to effectively future developments in scientific computing. We revised section 3 to make this point clearer.
>
> ---
>
> *W2 & W3 & Q3: Some test details are unclear, e.g., the experiment setup behind Figure 1 and the quantification of data shift in different cases. As the paper focuses on benchmarking the training methods, the details are essential for empirical analysis. The empirical results lack theoretical support. [...] Is there any theoretical analysis behind it to support the performance so that it can be generally applicable?*
>
> Thank you for the suggestion, it inspired a series of theoretical considerations which we added to the general response. We believe that these provide a new viewpoint that help interpret our results.
>
> Additionally, we have added an attention-based U-Net on the KS system. Similar to FNOs, the attention mechanism in this architecture is designed to make the network learn an effective mapping based on both global as well as local feature structures. The test loss values are added to Appendix D of our paper. The results support our conclusions and align well with previously tested architectures, from which we conclude that the attention mechanism does not fundamentally change the behavior of unrolling.
>
> ---
>
> ### Questions
>
> *Q1: Figure 1 is intuitive, but how do you get the curves?*
>
> Figure 1 is based on the theoretical behavior of unrolling in chaotic systems. A detailed explanation of the behavior of data-shift, as well as gradient accuracies in NOG and WIG setups can be found in the general response. The same reasoning is added in the revised version of the paper.
>
> ---
>
> *Q2: For correction setup, physics priors are available. The later experiments include changing the prior’s accuracy to show the performance variation. With priors, can physics-informed NN be used to solve the problem? Then, how about comparing the three variants of unrolling training with SOTA physics-informed NN for solving PDE?*
>
> This is an interesting thought. Our setups require networks that can be deployed both in predictions and corrections. The latter use a (low-fidelity) numerical solver to compute a prior approximation of one timestep. The numerical baseline thus outputs space-time discrete solutions. As such, our problem definition is not directly aligned with classical PINNs, which strive to find continuous solutions to PDE problems, and also utilize a continuously defined physics-informed loss. In theory, one could imagine a PINN being trained on the residual term of unclosed PDEs that results from coarse-graining or filtering the full solution. However, we are not aware of an existing framework that implements this approach. Similarly, it is unclear how unrolling could be integrated into this hypothetical PINN training. For these reasons, we consider the deployments of PINNs in our framework to be an interesting topic for future studies.
>
> ---
>
> *Q4: The training of multi-step unrolling seems to be more effective than one-step unrolling. How should the step m be chosen? Figure 6 shows the results with different m values in the Kuramoto-Sivashinsky equation. How about the other systems? Will the step size be decided from validation?*
>
> This is an interesting question, as a general statement about the optimal unrolling length m can’t be made. As we have outlined in the theoretical analysis above, the data shift is reduced for larger m. However, the rate of this trend largely depends on the underlying physics and the samples in the dataset. Similarly, the gradient instabilities and their rate of divergence depends on the underlying physics as well as the chosen discrete timestep. As the review correctly points out, this is studied for the KS system, but can also be seen for increasing m in the curriculum studies on the KOLM system. In practice, we recommend curriculum learning where the m is sequentially increased at training time. This can be seen as stepping through plot in Figure 6 from left to right. This procedure can be stopped once no benefits from longer unrollings are gained.

---

### Official Review · Reviewer_GanK · 2023-11-09

**Soundness:** 2 fair
**Presentation:** 2 fair
**Contribution:** 2 fair
**Rating:** 3
**Confidence:** 4

**Summary:**

This work studies the effect of different training strategies for predicting the long-term dynamics of physical systems. Specifically, three approaches are presented, namely, training for only one step (ONE) of unrolled trajectory, training on a trajectory with no gradient backpropagation throughout (NOG) due to non-availability of a differentiable solver, and training with gradient (WIG) using a differentiable solver. Several experiments are conducted on four different systems in terms of different architectures, system sizes, and tasks. Overall, the work is a benchmarking and dataset track work, although the authors have not mentioned it as the primary area.

**Strengths:**

* S1: Several useful insights are provided, such as training on a longer horizon trajectory is better, differentiable solvers provide improved accuracy in comparison to their non-differentiable counterparts, curriculum learning helps, and a medium-sized architecture is better.

* S2: Experiments are conducted on several complex and non-trivial systems, albeit on a smaller size (four) for benchmarking and dataset track.

**Weaknesses:**

* W1: Codes are not made available, and there is no reproducibility statement. Thus, it is unclear if the work will be reproducible, which is not congruent with the ICLR author guide.

* W2: The details of implementation are not completely mentioned in terms of how the differentiability of the solver was ensured. Was it implemented in PyTorch or JAX? Considering that the main contribution of the work is to understand the effect of training strategy based on differentiable and non-differentiable approaches, these details are important.

* W3: The statistical significance of the results is not clear. In many cases, the error bars are overlapping while claims on one method being better than the other are made. Statistical significance tests such as paired T-tests might be required to validate the claims.

* W4: Many of the claims in the manuscript are fairly trivial or shown in previous works. (1) Large networks yield better results. (2) Learning on trajectory is better (a large family of works on Lagrangian and Hamiltonian Neural Networks have already implemented this and demonstrated it. For example, see https://proceedings.neurips.cc/paper/2020/hash/9f655cc8884fda7ad6d8a6fb15cc001e-Abstract.html ) where they have also used a horizon of 4 timesteps while training the systems, (3) differentiable solvers are better for training ML models of dynamical systems (there is a family of works on JAX and specifically JAX-MD which demonstrates this extensively).

* W5: The experiments are performed on an extremely limited number of datasets and architectures. For a benchmarking and dataset paper, more extensive studies are expected, especially in a dense area where a large number of models and datasets are available. The architectures used are not state of the art (see questions for details).

**Questions:**

1. In Figure 2, both for prediction and correction, the same $f_\theta$ is shown. The same applies to the text as well. It is unclear how the same $f_{\theta}$ can act as predictor and corrector. The  $f_{\theta}$ in predictor takes $u^t$ and predicts $u^{t+1}$. Whereas the  $f_{\theta}$ in corrector takes $\hat{u}^{t+1}$ predicted by the physics-based solver and corrects it to  $u^{t+1}$. How can both be performed by the same $f_\theta$? More explanation, perhaps with the help of an example solver $\mathcal{S}$, can give some clarity.

2. It is mentioned that across architectures and network sizes, WIG has the lowest error. From the results, it seems to not be clear statistically as the error bars are mostly overlapping. A paired t-test or so may be conducted to establish the statistical significance of the result.

3. In Fig. 4, NOG is giving poorer results than ONE for large network sizes (0.2 M and 1.0M). Why so? Moreover, this is not aligned with the claims in the manuscript in the section "Disentangling Contributions".

4. There are two approaches for predicting the long-term dynamics of systems known as Deep Koopman operators, and Fourier Neural Operators. Both of them have shown superior performance in predicting the long-term dynamics of physical and chaotic systems. Especially, FNO with vision transformers has shown promise in predicting extremely complex systems, such as weather forecast at a planetary scale. For benchmarking work, it is important to consider the SOTA models. Moreover, such models may raise additional questions, as outlined in the next question.

5. In the context of neural operators and transformer architectures (which employ all pair attention), do the conclusions made in the paper still hold true? This is important as the SOTA
models rely on these architectures, and commenting on the applicability of these approaches toward the newer architectures is important.

---

> ### Author Response · Authors · 2023-11-20
> **Response to Reviewer GanK - Part 1/3**
>
> Thank you for the review. Please find individual responses to all of your concerns and questions below.
>
> ### Weaknesses
>
> *W1: Codes are not made available, and there is no reproducibility statement. Thus, it is unclear if the work will be reproducible, which is not congruent with the ICLR author guide*
>
>
> We want to emphasize that all of our code will be made available on github upon acceptance. This facilitates the easy reproduction of our results, as well as the utilization of all underlying differentiable solvers for future studies. Similarly, our datasets will be made available. These arrangements are crucial in light of the goal of our paper, where we aim to establish a platform of comparison that spans physical problems, numerical solvers, machine learning architectures as well as a large range of network sizes.
>
> ---
>
> *W2: The details of implementation are not completely mentioned in terms of how the differentiability of the solver was ensured. Was it implemented in PyTorch or JAX? Considering that the main contribution of the work is to understand the effect of training strategy based on differentiable and non-differentiable approaches, these details are important.*
>
> Our results were mostly computed with PyTorch, including the differentiable solvers. Only the Kolmogorov Flow setup used Tensorflow. We want to stress that, by definition, algorithmic differentiation is identical to machine precision for the same operations in these frameworks. To clarify these details, we have added this information to the setup descriptions in Appendix C.
>
> ---
>
> *W3 & Q2: The statistical significance of the results is not clear. In many cases, the error bars are overlapping while claims on one method being better than the other are made. Statistical significance tests such as paired T-tests might be required to validate the claims.*
>
> We thank the reviewer for the suggestion. To improve the interpretability of our results, a series of significance tests were performed on our trained models. We used Welch’s t-test with one-sided p-value calculation and compared NOG and WIG distributions to the ONE baseline. The resulting p-values are reported in the updated appendix of the paper. The differences in the studied training modalities are statistically significant for smaller network setups. As model sizes increase, the distribution of trained models becomes more similar. Our heavily overparameterized networks (e.g. 1.0M parameters for 48 degree of freedom KS system) all show highly accurate and stable predictions (see divergence time metric, Figure 20 Appendix C). Based on the theoretical considerations added in the general comment, this means that the attractor of the learned dynamics is closely aligned with the ground truth system. These setups thus display less data shift and the benefits of unrolling are reduced. When these conditions in the overparameterized regime are met, NOG models are at a disadvantage due to their mismatch between gradients and loss landscape. As discussed in our answer to W4, the heavily oversized architectures are not practically relevant for scientific computing due to their weak scaling compared to numerical approaches. This evaluation confirms that for relevant, small to medium sized networks our results are statistically significant and hence that conclusions can be drawn.
>
> ---
>
> *W4.1: Many of the claims in the manuscript are fairly trivial or shown in previous works. (1) Large networks yield better results. [...]*
>
> The observation that larger networks yield better results might seem trivial. However, our results allow insight beyond that fact: We provide convergence rates of the inference error with respect to the parameter count for machine-learned correction and prediction methods. For the physical systems, numerical solvers, and network architectures studied, these convergence rates are small in comparison to numerical approaches. A grid refinement of the underlying solver might thus yield better results than an extension of the neural network parameter space for the same additional computational burden. Based on this observation, we conclude that slimmer architectures are of greater interest to the scientific computing community.
> We believe that this is an important and novel consideration that goes beyond previous studies.
>
> We’d also like to point out that, unfortunately, many current submissions in the field (and also in this year’s ICLR) do not respect the seemingly obvious fact above: they compare models with strongly differing parameter counts in a single graph or table. We hope that stating and illustrating this effect clearly raises awareness in the community.

---

> ### Author Response · Authors · 2023-11-20
> **Response to Reviewer GanK - Part 2/3**
>
> ### Weaknesses
>
> *W4.2 & W4.3: Many of the claims in the manuscript are fairly trivial or shown in previous works. [...] (2) Learning on trajectory is better (a large family of works on Lagrangian and Hamiltonian Neural Networks have already implemented this and demonstrated it. For example, see [...] where they have also used a horizon of 4 timesteps while training the systems, (3) differentiable solvers are better for training ML models of dynamical systems [...].*
>
> Differentiable trajectory unrolling in itself is of course not new, and various papers have demonstrated the benefits of this approach. However, differentiability is generally not given in codebases used for scientific computing. These codebases are the product of long development and validation processes, where establishing differentiability would be a tremendous effort. The no-gradient setup provides a very attractive strategy that would only require an interface between a numerical simulator and the machine learning augmentation. Reliable statistical evaluations are necessary to estimate the efficacy of these training modalities and to steer future developments in scientific computing.
>
> ---
>
> *W5: The experiments are performed on an extremely limited number of datasets and architectures. For a benchmarking and dataset paper, more extensive studies are expected, especially in a dense area where a large number of models and datasets are available. [...]*
>
> In contrast to many previous studies, our work focuses on correction tasks and deploys a comparison of differentiable and non-differentiable training approaches. As such, a differentiable solver is at the base of each of our 3 main datasets. This stands in stark contrast to other studies (e.g. Stachenfeld et al. 2022) which are limited to predictive setups. For these, mostly existing or commercial solvers were used. In contrast, our work combines popular dataset choices in machine learning on differentiable solvers (e.g. Um et al. 2019, Kochkov et al. 2021), and thus constitutes a valuable asset for training and testing architectures based on these differentiable solvers.
>
> We trained a broad range of architectures that cover more than 3 orders of magnitude in network size, resulting in over 2000 trained models for this study. As further elaborated upon in the paper and our answer to W4, this capacity dimension of neural networks is often overlooked, but of critical importance to scientific computing. The range of networks covers common choices in prominent previous work. For instance, the 1.0M parameter graph-network aligns with Branstetter et al. (2022), while the convolutional network with 250k parameters from Kochkov et al. (2021) is also contained in our capacity range. Similarly, the convolutional network sizes (60k and 250k) studied by Um et al. (2020) fall within the same range. This is also true for the convolutional ResNet architectures in Stachenfeld et al. (2022) with 200k parameters for 1D problems and 580k parameters in 2D.
>
> ***
>
> ### Questions
>
> *Q1: In Figure 2, both for prediction and correction, the same is shown. The same applies to the text as well. It is unclear how the same can act as predictor and corrector. The in predictor takes and predicts. Whereas the in corrector takes predicted by the physics-based solver and corrects it to. How can both be performed by the same?*
>
> We introduced this notation for brevity, but see how sharing the same function for correction and prediction tasks can initially lead to confusion. For a given network baseline (e.g. CNNs and graph nets) and parameter counts, the resulting architecture is indeed identical. Similarly, the input and output spaces of the networks are identical, as they are simply the discrete vector fields before and after correction or prediction. However, as the reviewer correctly pointed out, the learned mapping is different for correction and prediction tasks, and the actual weights and biases are trained individually for correction and prediction. Crucially, the underlying architectures are identical. One key point of our work is to show that a transition between prediction and correction is possible. Our results can be used to estimate the potential gains when adding a (low fidelity) numerical solver to the learning task, while we also provide concrete results for the three main training modalities that are relevant in this setting. Thank you for pointing out this issue. We have revised the paper to include these motivating thoughts and made the notation clearer.

---

> ### Author Response · Authors · 2023-11-20
> **Response to Reviewer GanK - Part 3/3**
>
> ### Questions
>
> *Q3: In Fig. 4, NOG is giving poorer results than ONE for large network sizes (0.2 M and 1.0M). Why so? Moreover, this is not aligned with the claims in the manuscript in the section "Disentangling Contributions".*
>
> We agree that an explanation for this was not readily apparent in our initial submission. Based on your and the other reviewers’ feedback we have now included a theoretical analysis in the paper and in our main response. We believe that it can help establish an intuition for unrolled setups. This is also applicable to this specific question. The largest models achieve dynamics that closely match the state space of the ground truth dynamics’ attractor. When the learned dynamics become increasingly similar to the ground truth, unrolling may not be necessary to expose the full learned dynamics. While unrolling now offers less benefit, NOG still maintains a mismatch between the calculated gradients and the underlying loss landscape. This could ultimately hurt performance. WIG does not have this disadvantage as gradients accurately match the loss, and the unrolled steps lie within the stable horizon, avoiding gradient explosion.
>
> ---
>
> *Q4 & Q5: There are two approaches for predicting the long-term dynamics of systems known as Deep Koopman operators, and Fourier Neural Operators. Both of them have shown superior performance in predicting the long-term dynamics of physical and chaotic systems. [...] For benchmarking work, it is important to consider the SOTA models. [...] In the context of neural operators and transformer architectures (which employ all pair attention), do the conclusions made in the paper still hold true? This is important as the SOTA models rely on these architectures, and commenting on the applicability of these approaches toward the newer architectures is important*
>
> Of course, there are exciting developments on the architecture side that could ultimately improve the accuracy of long-term correction and prediction networks. However, the underlying architecture of these networks is not at the core of our work, and we also do not present these architectures necessarily as the best-performing choice. The chosen architectures are, however, a fundamental building block of more recent developments and still serve as a basis of comparison in related work. The goal of our study is to establish a guide for designing training procedures that yield the best results for a given network. Given the scale of our empirical study across architectures, solvers, physical systems, and network sizes, as well as the added theoretical analysis, we are confident that our results translate to other architectures. To address your skepticism on this issue, we have trained an attention-based U-Net on the KS system. Similar to FNOs, the attention mechanism in this architecture is designed to make the network learn an effective mapping based on both global as well as local feature structures. The test loss values are added to Appendix D of our paper. The results support our conclusions and align well with previously tested architectures, from which we conclude that the attention mechanism does not fundamentally change the behavior of unrolling.
>
> For Koopman operators, we are skeptical whether a separate Koopman embedding would really be beneficial in a corrective setup, which fundamentally requires communication between the network and a numerical solver at each timestep in real space. While the integration of this architecture in correction setups would surely be interesting, we'd prefer to leave this development for future work.

---

> > ### Comment · Reviewer_GanK · 2023-12-02
> > **Thank you**
> >
> > I thank the authors for their response to the comments. However, some of the concerns still remain unaddressed. Accordingly, I maintain the score.

---

### Author Response · Authors · 2023-11-20
**General response to all reviewers - Part 1/2**

## General Response

Thanks to all reviewers for their detailed comments, questions and constructive suggestions. They have inspired a series of changes, and we think they have helped us fundamentally improve the completeness of the our work.

We would like to share some theoretical insights in this general response, as they will prove to be useful in addressing some of the individual questions. Additionally, they help establishing an intuition for unrolled chaotic systems that ultimately helps us interpret the presented results.

### Theoretical analysis 1/2
Suppose we study a chaotic dynamical system

$$\tilde u_{n+1} = f(\tilde{u}_{n},\nabla \tilde{u}, \nabla^2 \tilde {u},  . . .),$$

where $\tilde{u}$ represents a discrete ground-truth state. The chaotic dynamics drive this system to an attractor $A_f$ representing a subset of the phase space of $\tilde{u}$.
In prediction and correction tasks, a neural simulator learns the dynamics

$$u_{n+1} = f_\theta(u_{n},\nabla u, \nabla^2 u . . . ),$$

which approximates the evolution of the ground truth system. Since perfect reproduction of $f$ is generally not achieved, differences between the trajectories of $\tilde{u}$ and $u$ exist, leading to $A_f \neq A_{f_\theta}$.
Crucially, there is no guarantee for ONE-step training that the states $u$ observed during training sufficiently represent $A_{f_\theta}$. In other words,

$$\forall \tilde u_i  \in A_f, A_{f_\theta}:  \lim_{N\rightarrow\infty} [ f_\theta(\tilde u_0), f_\theta(\tilde u_1), . . . , f_\theta(\tilde u_N) ] \neq A_{f_\theta}.$$

This means that we are not guaranteed to explore the attractor of the learned system $A_{f_\theta}$ when only observing states based on one discrete evolution of $f_\theta$, regardless of the dataset size $N$.
Let us now suppose we unroll $m$ steps during training such that
$ u^m = f_\theta^m(u)$,
where $f_\theta^m$ denotes the autoregressive evolution of $m$ steps. Based on the definition of the attractor $A_{f_\theta}$ we can state that

$$\lim_{m\rightarrow\infty}[f_\theta^0(\tilde{u}), f_\theta^1(\tilde{u}), . . . , f_\theta^m(\tilde{u})] = A_{f_\theta}.$$

In contrast to ONE-step training, unrolled training thus exposes the inference attractor $A_{f_\theta}$ at training time for sufficiently large $m$.
Precisely the difference between the observed training samples and the inference attractor $A_{f_\theta}$ is commonly known as data shift in machine learning. The dark red curve in Figure 1 shows how this data shift is thus reduced by choosing larger $m$.

---

> ### Author Response · Authors · 2023-11-20
> **General response to all reviewers - Part 2/2**
>
> ### Theoretical analysis 2/2
>
> Let us now study the gradients of the NOG unrolling variant. In Table 1 and Appendix A of our paper we derived the precise gradient equations for NOG and WIG setups. These are
>
> $$
> \begin{equation}
> \frac{\partial \mathcal{L}_{NOG}}{\partial \theta}=\sum _{s=1}^m \frac{ \partial \mathcal{L}^s }{\partial f _\theta^s}\frac{\partial f _\theta^s}{\partial \theta}, \quad \quad
> \frac{\partial \mathcal{L} _{WIG}}{\partial \theta} =\sum _{s=1}^m \sum _{B=1}^s \bigg[ \frac{\partial \mathcal{L} ^b}{\partial f _\theta^b}\bigg ( \prod _{b=s}^{B+1} \frac{\partial f _\theta^{b}}{\partial f _\theta^{b-1}}\bigg) \frac{\partial f _\theta^B}{\partial \theta} \bigg].
> \end{equation}
> $$
>
> Note that WIG unrolling calculates the true gradient. We can thus derive the gradient inaccuracy of the NOG setup as
>
> $$
> \begin{equation}
> \frac{\partial \mathcal{L}_{WIG}}{\partial \theta} - \frac{\partial \mathcal{L} _{NOG}}{\partial \theta} =\sum _{s=1}^m  \sum _{B=1}^{s-1} \bigg[\frac{\partial \mathcal{L}^b}{\partial f _\theta^b} \bigg( \prod _{b=s-1}^{B+1}\frac{\partial f _\theta^{b}}{\partial f _\theta^{b-1}} \bigg) \frac{\partial f _\theta^B}{\partial \theta} \bigg] .
> \end{equation}
> $$
>
> We can now integrate a property of chaotic dynamics derived by Mikhaeil et al. (2022). Herein, the authors show that for chaotic systems the Jacobians
> $\mathbf{J_s}=\dfrac{\partial f_\theta^{s}}{\partial f_\theta^{s-1}}$
> have eigenvalues larger than 1 in the geometric mean. Thus,
>
> $$\bigg|\bigg|\prod_{b=s-1}^{B+1}\frac{\partial f_\theta^{b}}{\partial f_\theta^{b-1}}\bigg|\bigg| > 1$$
>
> holds in this case.
> This means that the gradient inaccuracy increases with the unrolling. We can also observe that the NOG gradient inaccuracy grows with $\propto m^2$, while the NOG gradients only linearly depend on $m$. As a consequence, the gradients computed in the NOG system diverge from the true (WIG) gradients for increasing $m$, as shown with the blue line in Figure 1. The gradient approximations used in NOG thus do not accurately match the loss used in training. This hinders the network optimization.
>
> Let us finally consider the full gradients of the WIG setup themselves. We can use Theorem 2 from Mikhaeli et al. (2022) which states that
>
> $$\lim_{m\rightarrow\infty}\big|\big|\frac{\partial f_\theta^m}{\partial f_\theta^0}\big|\big|=\infty,$$
>
> for almost all points on $A_{f_\theta}$. The gradients of the WIG setup explode exponentially for long horizons $m$. For all commonly chosen time step size, the WIG setup gradient explode less quickly than those of NOG, as indicated by the orange line below the blue one in Figure 1.
> As a direct consequence, the training of the WIG setup becomes unstable for chaotic systems when $m$ grows to infinity.
>
> We can summarize the above findings as follows. Unrolling training trajectories for data-driven learning of chaotic systems reduces the data-shift, as the observed training samples converge to the learned attractor. At the same time, long unrollings lead to unfavorable gradients for both NOG and WIG setups. A range of medium-sized unrolling horizons might exist, where the benefits of reducing the data-shift outweigh instabilities in the gradient. These setups are studied in the paper and exposed in Figure 6 for the KS system.
>
> ### Changes to the paper
>
> Inspired by the reviewers' feedback, we are in the process of reflecting the comments and answers in our submission.
> Alongside the addition of a test for statistical significance, the above theoretical consideration constitute the major changes to the paper. Additionally, some sections are reworked to clarify the reviewers' concerns.
> An updated version of the PDF will follow shortly.

---

### Author Response · Authors · 2023-11-22
**Revised PDF is now online**

Dear reviewers,

The revised PDF is now online!
We have included all changes as announced in the general or individual responses.
The new or edited sections are color-coded in blue.

Thanks again for the comments.

---

### Meta-Review · Area_Chair_Vk8S · 2023-12-05

**Metareview:**

This paper examines the impact of unrolling strategies during training on the predicting of the long-term dynamics of physical systems. The reviewers found strength in a number of aspects of the paper, for example the scaling trends and the reasonably good performance of methods without temporal unrolling. For the most part, the writing and presentation were clear, but it was noted that a number of important details regarding the algorithms were not presented, making it difficult to fully understand and evaluate some aspects of the proposed methods. Furthermore, there were concerns that the evaluations were unconvincing, the comparisons with related work were insufficient. and the main take-aways were unclear. Further refinement of the presentation, including improved messaging of the potential impact and better demonstrations that that impact could be achieved would significantly improve the manuscript.

**Justification For Why Not Higher Score:**

Unconvincing experimental evaluations, imprecise description of some algorithmic details, and unclear practical take-aways.

**Justification For Why Not Lower Score:**

N/A

---

### Decision · Program_Chairs · 2024-01-16

Reject